# Mercury in the Arctic tundra snowpack: temporal and spatial concentration patterns and trace-gas exchanges

Yannick Agnan[1,2], Thomas A. Douglas[3], Detlev Helmig[4], Jacques Hueber[4], Daniel Obrist[5,2]

[1]Sorbonne Université, CNRS, EPHE, UMR Metis, F-75252, Paris, France
[2]Division of Atmospheric Sciences, Desert Research Institute, Reno, Nevada 89523, USA
[3]US Army Cold Regions Research and Engineering Laboratory, PO Box 35170, Fort Wainwright, Alaska 99709, USA
[4]Institute of Arctic and Alpine Research, University of Colorado, Boulder, Colorado 80309, USA
[5]Department of Environmental, Earth, and Atmospheric Sciences, University of Massachusetts, Lowell, MA 01854, USA

*Correspondence to*: Yannick Agnan (yannick.agnan@biogeochimie.fr) and Daniel Obrist (daniel_obrist@uml.edu)

**Abstract.** In the Arctic, the snowpack forms the major interface between atmospheric and terrestrial cycling of mercury (Hg), a global pollutant. We investigated Hg dynamics in an interior arctic tundra snowpack in northern Alaska during two snow seasons. Using a snow tower system to monitor Hg trace gas exchange, we observed consistent concentration declines of gaseous elemental Hg ($Hg^0_{gas}$) from the atmosphere to the snowpack to soils. The snowpack itself was unlikely a direct sink for atmospheric $Hg^0_{gas}$. In addition, there was no evidence of photochemical reduction of $Hg^{II}$ to $Hg^0_{gas}$ in the tundra snowpack, with the exception of short periods during late winter in the uppermost snow layer. The patterns in this interior arctic snowpack hence differ substantially from observations in arctic coastal and temperate snowpacks. We consistently measured low concentrations of both total and dissolved Hg in snowpack throughout the two seasons. Chemical tracers showed that Hg was mainly associated with local mineral dust and regional marine sea spray inputs. Mass balance calculations show that the snowpack represents a small reservoir of Hg, resulting in low inputs during snowmelt. Taken together, the results from this study suggest that interior arctic snowpacks are negligible sources of Hg in the Arctic.

## 1 Introduction

Mercury (Hg) is a neurotoxic pollutant of worldwide importance that is transported over long distances in the atmosphere as gaseous elemental Hg ($Hg^0_{gas}$), and thus reaches remote environments (Cobbett et al., 2007; Driscoll et al., 2013; Sprovieri et al., 2010). In the Arctic, modern atmospheric Hg deposition has increased about three-fold from pre-industrialized background levels (Fitzgerald et al., 2005), similar to increases observed in temperate locations, although other studies suggest much stronger increases (e.g., Enrico et al., 2017). The increase in Hg loading has led to vulnerability of polar ecosystems to Hg contamination due to detrimental impacts to wildlife and humans, in particular through biomagnification processes across trophic levels (Atwell et al., 1998).

Representing about 26% of the global land surface area, polar regions are unique environments with specific physical, chemical, and biological processes affecting pollutant cycles including that of Hg (Douglas et al., 2012). In particular, most of

the northern latitudes are covered by a laterally continuous snowpack during long periods of the year. In the Alaskan tundra, the surface snow cover is present about two thirds of the year (Cherry et al., 2014). The snowpack hence forms a critical interface between the arctic atmosphere, tundra ecosystems, and underlying tundra soils. Trace gas exchanges between the atmosphere and the tundra are modulated by sinks and sources below and within snowpack, by snow diffusivity, snow height, and snow porosity (Dominé and Shepson, 2002; Lalonde et al., 2002; Monson et al., 2006). The snowpack accumulates nutrients, pollutants, and impurities that are deposited by snowfall and dry deposition processes, all of which can subsequently be transported to underlying ecosystems during snowmelt (Bergin et al., 1995; Uematsu et al., 2000).

The snowpack plays an important role for the cycling of Hg as well, including for atmospheric deposition, photochemical redox reactions, and associated phase changes between solid and gaseous Hg that can volatilize Hg to the atmosphere (Douglas et al., 2008, 2012; Faïn et al., 2013; Mann et al., 2014; Steffen et al., 2013). In particular, temperate and arctic studies have shown that the snowpack can serve as sink or source of $Hg^0_{gas}$, whereby photochemical reduction of snow-bound $Hg^{II}$ can produce $Hg^0_{gas}$, and oxidation processes can reversely scavenge atmospheric $Hg^0_{gas}$ in snow (Faïn et al., 2013; Lalonde et al., 2002; Mann et al., 2011). Photochemical reactions occur primarily in the top 10 cm of the snowpack, where sunlight radiation transmits and is absorbed and scattered by snow crystals (Faïn et al., 2007; King and Simpson, 2001). The degree of photochemical production of $Hg^0_{gas}$ and subsequent atmospheric re-volatilization from the snowpack can be significant, as shown in temperate snowpacks with strong recurring daytime atmospheric emissions of $Hg^0_{gas}$ throughout the winter season (Faïn et al., 2013). In global models, snowpack $Hg^0_{gas}$ emissions can account for ~50% of all snowpack Hg (Corbitt et al., 2011). The reverse process—oxidation of $Hg^0_{gas}$ to $Hg^{II}$—, has also been proposed to occur in the dark snowpack deeper in the snow profile (Faïn et al., 2007, 2013; Mann et al., 2015), resulting in concentration declines of $Hg^0_{gas}$ with depth in the snowpack. To our knowledge, no direct in situ measurement of snowpack $Hg^0_{gas}$ dynamics, however, is available in the field in the interior arctic snowpack.

In the Arctic and Antarctic, Hg cycling is also affected by atmospheric Hg depletion events (AMDEs), which are observed primarily in the springtime along coastal locations (Angot et al., 2016a; Dommergue et al., 2010; Schroeder et al., 1998; Steffen et al., 2008). During AMDEs, atmospheric $Hg^0_{gas}$ concentrations fluctuate strongly due to atmospheric conversion of $Hg^0_{gas}$ to oxidized $Hg^{II}$. Because $Hg^{II}$ is subject to faster deposition (Schroeder and Munthe, 1998; Selin, 2009), AMDEs result in Hg temporarily deposited from the atmosphere to the arctic ecosystems. AMDEs are considered to be initiated by halogens (Brooks et al., 2008; Obrist et al., 2011; Steffen et al., 2008), such as bromine and chlorine radicals released from sea salt by photochemical processes (Simpson et al., 2007). AMDEs have been mainly observed along the coasts, e.g., at Barrow in Alaska (Douglas et al., 2008), Alert in Canada (Steffen et al., 2002), Ny-Ålesund in Svalbard (Ferrari et al., 2008), McMurdo in Antarctica (Brooks et al., 2008), as well as directly over the sea ice (Moore et al., 2014; Nerentorp Mastromonaco et al., 2016). The impacts of AMDEs at inland sites is reduced with increasing distance from the coast (Douglas and Sturm, 2004; Obrist et al., 2017; Van Dam et al., 2013).

The objectives of this study were to characterize Hg dynamics in the inland arctic snowpack at Toolik Field Station, and along a 170-km transect between this site and the arctic coast. For the first time, we comprehensively linked trace gas fluxes of $Hg^0_{gas}$

in interstitial snow air with the seasonal development of total Hg ($Hg_{tot}$) and dissolved Hg ($Hg_{diss}$) bound in the snowpack to assess conversions between volatile and solid Hg in the arctic snowpack. We specifically aimed to assess: (1) temporal and vertical $Hg^0_{gas}$ patterns to quantify exchanges of $Hg^0_{gas}$ in the atmosphere–snowpack–soil continuum; (2) impacts of springtime AMDEs on snowpack Hg deposition to and emission from the inland arctic snowpack; (3) temporal and vertical concentration and mass patterns of the snowpack $Hg_{tot}$ and $Hg_{diss}$, to estimate Hg deposition throughout the snow accumulation period and pool of Hg available through snow melt; and (4) relationships of snow Hg concentrations with major ion concentrations and oxygen and hydrogen stable isotopes in precipitation to determine potential origins of Hg contained in the snowpack.

## 2 Materials and methods

### 2.1 Study site

Measurements were mainly performed at Toolik Field Station (Alaska, USA) over two full snow cover seasons from October 2014 to May 2016. The research station is located on the north slopes of the Brooks Range (68° 38' N, 149° 36' W) at an elevation of 720 m a.s.l, approximately 200 km south of the Arctic Ocean (Fig. 1, orange bullet). The area is characterized by gently sloping hills comprised of poorly drained silty loams underlain by continuous permafrost 250–300 m deep (Barker et al., 2014). Lithology is characterized by glacial till over Cretaceous sedimentary substrates (shale, claystone, siltstone, and sandstone; Alaska Division of Oil and Gas, 2008). The ecotype is classified as an acidic tussock tundra (Shaver and Chapin, 1991) with vegetation composed of scrubby plants (e.g., *Cassiope tetragona* (L.) D.Don, *Arctostaphylos alpinus* (L.) Spreng.), shrubs (e.g., *Betula nana* L., *Salix pulchra* Cham.), tussock grasses (*Carex*), and a variety of mosses and lichens. The mean annual air temperature is −8.5 °C, and mean annual precipitation is 312 mm (Cherry et al., 2014). In the two measurement years, the tundra was covered by snow for 236 and 248 days (i.e., 65 and 68% of the year) in the 2014–2015 and 2015–2016 seasons, respectively.

Snowpack sampling was also performed along a transect between Toolik and the Arctic Ocean in March 2016 (Fig. 1, yellow bullets). Detailed geographical characteristics of the sample sites are given in Table S1. A total of eight study sites were sampled from south (500 m a.s.l.) to north (20 m a.s.l.). All the sampled sites were characterized by similar ecosystems and lithology (including undifferentiated volcanic Upper Tertiary beds to the north) as described above for the Toolik area.

### 2.2 Trace gas in the atmosphere, interstitial snow air, and soil pores

We continuously sampled and analyzed interstitial air of the tundra snowpack at Toolik using a snow tower (Fig. S1) as described in detail by Seok et al. (2009) and Faïn et al. (2013). In summary, a snow tower consists of an air inlet manifold placed in the snowpack, so sampling of trace gases can be remotely alternated between various snow depths for undisturbed sampling of interstitial snow air throughout an entire snow season. The snow tower used at Toolik consisted of six 60 cm aluminum cross arms mounted at heights of 0, 10, 20, 30, 40, and 110 cm above the ground surface. Gas inlets were mounted to each cross arm allowing vertical sampling of snow interstitial air for analysis for multiple trace gases, including $Hg^0_{gas}$, $CO_2$,

and $O_3$. Each cross arm supported a pair of air inlets fitted with 25 mm syringe filters with 1 μm glass fiber membranes (Pall Life Sciences, Ann Arbor, MI, USA). Perfluoroalkoxy Teflon® tubing with equal lengths (35 m) were directed in a heated conduit to solenoid valves in the laboratory that allowed for sequential sampling of trace gases at the six different snowpack heights. The snow tower was deployed over the tundra in August of each year prior to the onset of snowfall. When the snow tower was subsequently covered by the accumulating snowpack, this set-up allowed sequentially continuous sampling of snow interstitial air without any disturbance. Inlets were sampled sequentially, 10 min at a time (i.e., averages of two individual measurements of 5 min), resulting in a 60-min sampling cycle. Corresponding trace gas sampling was performed below the snowpack in tundra soils at depths of 10, 20, and 40 cm using Teflon® soil trace gas wells (Obrist et al., 2014, 2017). Both organic and mineral soil profiles were considered in this study, distant of 5 m as described in Obrist et al. (2017). Atmospheric air sampling was performed using the top snow tower air inlets which always were above the developing snowpack, as well as on a nearby micrometeorological tower at a height of 3.6 m above ground. All interstitial snow, soil pore, and atmospheric inlets were connected by Teflon® tubing and solenoid valves to trace gas monitors in a nearby (10–30 m distance) field laboratory that were operated year-round.

Gaseous $Hg^0$ concentrations were measured using two Tekran 2537B analyzers (Tekran Instruments Corporation, Toronto, ON, Canada), one shared for interstitial snow air and atmospheric measurements, and the other shared for soil gas and atmospheric measurements. The discrepancy in $Hg^0_{gas}$ measurements observed between the two Tekran instruments along the two seasons was on average 7%; concentration data showed here were adjusted using the differences between the two atmospheric inlets. Air sampling was alternated between different snowpack heights every 5 min so that a full sequence of air extraction from the snowpack (six inlet heights) was achieved every 30 min. Interstitial snow, soil pore, and atmospheric measurements continued through the entire winter with only small-time periods of interruptions due to power failures or other technical problems. Additional trace gases were measured along with $Hg^0_{gas}$, including concentrations of $CO_2$ using a LI-840A (LI-COR Inc., Lincoln, NE, USA).

## 2.3 Snow sampling and physical and chemical characterization

### 2.3.1 Snow sampling

At Toolik, we characterized Hg in the snowpack both over the undisturbed tundra and the adjacent frozen Toolik Lake (within 200 m of the tundra location). Two snow pits were sampled on five dates between October and May in the 2014–2015 season, and on four dates between December and June in 2015–2016. For each pit, we vertically excavated snow samples using a stainless-steel snow cutter (RIP 1 cutter 1000 cc), clean latex gloves, and trace metal Nasco Whirl-Pak® (The Aristotle Corporation, Stamford, CT, USA) HDPE plastic bags. We sampled at 10 cm-layer increments from the top to the bottom of the snowpack. Samples from two perpendicular walls of the pit were each pooled together per layer for analysis. Snow height, density, and temperature were measured for each layer, and frozen snow samples were stored in a cooler before transferring to a −20 °C freezer. Snow water equivalent (SWE), which represents the amount of water stored in the snowpack, was

calculated using snow density measurements in incremental 10 cm-layers, multiplied by snow height. Additional sampling of surface snow was performed over the tundra for a total of 17 sampling dates. The top 3 cm of the snowpack was collected in triplicate within a distance of 5 m into Nasco Whirl-Pak® plastic bags using clean latex gloves. Sampling along the south to north transect was performed over two days in March 2016.

## 2.3.2 Chemical analyses

In the laboratory, we melted snow samples overnight in the Nasco Whirl-Pak® bags at room temperature in the dark, and melted snow samples were subsequently analyzed for Hg. A fraction of snowmelt was directly transferred to 50 mL polypropylene tubes (Falcon®, Corning Incorporated, Corning, NY, USA) for analysis of $Hg_{tot}$. For $Hg_{diss}$, snowmelt water was filtered using 0.45 µm Acrodisc® filter with polyethersulfone membrane (Pall Corporation, Port Washington, NY, USA) into 50 mL Falcon® polypropylene tubes. In addition, filtered meltwater was used in 60 mL high-density polyethylene tubes (VWR®, Radnor, PA, USA) for determination of major cations, anions, and stable isotopes ($^2$H and $^{18}$O). Total Hg and $Hg_{diss}$ concentrations were determined using Tekran 2600 cold-vapor atomic fluorescence spectrometry (Tekran Instruments Corporation, Toronto, ON, Canada) using a bromine monochloride (BrCl) digestion and reduction by stannous chloride ($SnCl_2$) following EPA method 1631 (US EPA, 2002). The detection limits (DL), determined as 3-times the standard deviation of blank samples, averaged 0.08 ng L$^{-1}$. For statistic purpose, values below the DL were included as $0.5 \times DL$. Analyzer performance was determined by 5 ng L$^{-1}$ standards analyzed every 10 samples, and recovery averaged between 93 and 107%. Laboratory and field blanks were conducted, and we evaluated any potential metal contamination of the stainless-steel snow cutter by analyzing Milli-Q water in contact with the snow cutter; all these blank determinations were bellow detection limits. Major cation and anion concentrations were quantified at the U.S. Army Cold Regions Research and Engineering Laboratory's (CRREL) Alaska Geochemistry Laboratory on Fort Wainwright, Alaska, with a Dionex ICS-3000 ion chromatograph. An AS-19 anion column and a CS-12A cation column (Dionex Corporation Sunnyvale, California) were used, each with a 10 µL injection volume. A gradient method using potassium hydroxide (20 to 35 µmol L$^{-1}$) was used for anion analyses, while cation analyses used methane sulfonic acid eluent with a concentration of 25 µmol L$^{-1}$ in isocratic mode. The flow rate was 1 mL min$^{-1}$ and the operating temperature was 30 °C. The ion chromatograph was calibrated using standards with a range from 0.5 to 50 mg L$^{-1}$. Repeat analyses of calibration standards from 0.5 to 50 mg L$^{-1}$ yielded a precision of ±5%. Peaks were identified using Chromeleon (Dionex) and verified visually.

Stable isotopes of oxygen and hydrogen were also measured at CRREL Alaska using Wavelength-Scanned Cavity Ringdown Spectroscopy on a Picarro L2120i (Sunnyvale, California). Standards and samples were injected into the analyzer for seven separate analyses. Results from the first four injections were not used to calculate the stable isotope values to eliminate internal system memory. The mean value from the final three sample injections was used to calculate the mean and standard deviation value for each sample. Values are reported in standard per mil notation. Repeated analyses of five internal laboratory standards representing a range of values spanning the samples analyzed and analyses of SMOW, GISP, and SLAP standards

(International Atomic Energy Agency) were used to calibrate the analytical results. Based on thousands of these standards analyses and of sample duplicate analyses we estimate the precision is ±0.2‰ for $\delta^{18}O$ and ±0.5‰ for $\delta^2H$.

## 2.4 Data processing and statistical analyses

We performed all data processing and statistical analyses with RStudio 1.1.383 (RStudio Inc., Boston, Massachusetts, USA) using R 3.4.2 (R Foundation for Statistical Computing, Vienna, Austria). Averaged data and variance in figures and tables are shown as mean ± standard deviation. Significant differences were determined with the Kruskal-Wallis test ($\alpha = 0.05$). We performed plots with *ggplot2*, *ggtern*, and *lattice* R packages, and used normality (eq $L^{-1}$) for the ternary diagram. Geographical maps were prepared using Quantum GIS 2.18 (Quantum GIS Development Team, 2017).

## 3 Results and discussion

### 3.1 Snowpack development and snowpack physics

Due to high wind conditions in the Arctic tundra (Cherry et al., 2014), the physical development of the snowpack and its depth and the thickness of wind slab layers at Toolik were subject to significant drifts and changes in snowpack height, and were hence highly variable spatially and temporally throughout the winter season. The average snow height over the tundra site (shown in gray bars in Fig. 2) was continuously measured in both winters using a camera set to record daily pictures and using reference snow stakes placed in the snowpack. In the 2014–2015 season, the average snowpack height was 37 cm, with a standard deviation of 12 cm and a maximum depth of 60 cm. In the 2015–2016 season, the snowpack was almost half of that of the previous year, with an average snowpack height of 19 cm, a standard deviation of 7 cm, and a maximum depth of 35 cm. Based on snow pit measurements in the 2014–2015 season, we observed an increase of snow density with time, from an average of 0.18 g cm$^{-3}$ in October to 0.26 g cm$^{-3}$ in March (blue lines in Fig. 2). No clear temporal pattern was observed in the 2015–2016 season when average snow density; it ranged between 0.28 and 0.30 g cm$^{-3}$. Results showed similar temporal evolution as snow heights, with maximum SWE observed in March in both snow seasons of 158 and 116 mm, respectively. Snowpack temperatures were highly variable throughout the seasons and also strongly differed vertically within the snowpack (red lines in Fig. 2). Temperatures ranged from −34 to 0 °C in the top of the snowpack and from −21 to −1 °C in the bottom of the snowpack; temperatures showed strong increases from the top to the bottom of the snowpack, illustrating the important insulating function that the snowpack has in the cold Arctic winter and spring months. Minimum snowpack temperatures were recorded during the January 26th 2015 sampling event when air temperatures were −40 °C.

The snowpack over the adjacent frozen lake showed an average density of 0.23 g cm$^{-3}$ and temperatures ranged between −18 and 0 °C. The snow height over Toolik Lake was much lower than that over the tundra, with snow heights consistently <15 cm for both seasons. The maximum SWE calculated above the lake was 40 and 42 mm for the two snow seasons, respectively.

The transect between Toolik and the Arctic Ocean performed in March 2016 showed snowpack height ranging between 30 and 66 cm. The maximum height was observed at one site located 55 km from the Arctic Ocean where presence of dense

shrubs up to 40 cm height induced accumulation of local drifting snow due to high roughness. Snow density (between 0.19 and 0.26 g cm$^{-3}$) and temperatures (between −20 and −10 °C) followed the same trends as observed at Toolik with decreasing density and increasing temperatures with snowpack thickness. The calculated SWE averaged 104 mm and ranged between 70 and 164 mm.

## 3.2 Gaseous Hg$^0$ in the atmosphere–snowpack–soil continuum

### 3.2.1 Gaseous Hg$^0$ concentration profiles

Gaseous Hg$^0$ concentrations were measured at Toolik over two years in the atmosphere, in snowpack interstitial air at up to five inlet heights, and in soil pore air in the tundra ecosystem. Data coverage was 183 and 207 days for the 2014–2015 and 2015–2016 seasons, respectively, with only few periods when system failures resulted in lack of data. A continuous temporal record of the Hg$^0_{gas}$ concentration profile in the snowpack is presented in Fig. 3a for the 2014–2015 season, i.e., when the snowpack was deeper compared to the 2015–2016 season, and compared to a similar record from a temperate snowpack based on published data (Fig. 3b; Faïn et al., 2013; note different y-scale of figure panels). In addition, full time-averaged atmosphere–snowpack–soil Hg$^0_{gas}$ diffusion profiles are shown for the entire two winter seasons: 2014–2015 (Fig. 4a–c) and 2015–2016 (Fig. 4d–f). Gaseous Hg$^0$ concentrations were averaged for each season for three different periods, i.e., November to December (representing early winter and full darkness), January to February (representing mid-winter and full darkness), and March to April (when sunlight emerged and when occasional AMDEs were active). Note that standard deviations indicate natural fluctuations in Hg$^0_{gas}$ concentrations as observed in Obrist et al. (2017).

The Hg$^0_{gas}$ measurements consistently showed strong concentration gradients in the atmosphere–snowpack–soil continuum with highest concentrations in the atmosphere (on average, $1.18 \pm 0.13$ and $1.09 \pm 0.13$ ng m$^{-3}$, respectively) and lowest concentrations in soils (often below detection limits of 0.10 ng m$^{-3}$). This pattern was consistent over two independent soil profiles measured at this site, one mainly consisting of organic soils and one soil profile dominated by mineral soil horizons. Hg$^0_{gas}$ concentrations in the snowpack were between concentration in the atmosphere and in soils, and showed pronounced patterns of decreasing concentrations from the top to the bottom of the snow profile. In the first year, Hg$^0_{gas}$ concentrations decreased from the top snowpack inlet (i.e., 40 cm above the ground; average Hg$^0_{gas}$ concentration of 1.18 ng m$^{-3}$) to the lower snowpack sampling heights (30, 20, and 10 cm above the ground; average Hg$^0_{gas}$ concentrations of 1.11, 1.00, and 0.76 ng m$^{-3}$, respectively), and showed the lowest Hg$^0_{gas}$ concentrations at the soil–snowpack interface (0 cm: 0.53 ng m$^{-3}$). Due to a much shallower snowpack in the 2015–2016 season and an absence of measurements at 0 cm height due to line freezing of the lowest inlet, the profile of Hg$^0_{gas}$ was less pronounced compared to 2014–2015. However, we similarly found a Hg$^0_{gas}$ decline from upper to lower snowpack heights (e.g., Hg$^0_{gas}$ concentrations of 1.09 ng m$^{-3}$ in the atmosphere, 1.02 ng m$^{-3}$ at 20 cm, and 0.88 ng m$^{-3}$ at 10 cm height above ground). In a previous paper, we reported a small rate of continuous Hg$^0_{gas}$ deposition from the atmosphere to the tundra—measured by a micrometeorological tower—during much of the snow-covered season, with the exception of short time periods in spring when AMDEs occurred at Toolik (Obrist et al., 2017). Here, we show that these flux

measurements are supported by consistent $Hg^0_{gas}$ concentration gradients that existed through both seasons and that showed that snowpack $Hg^0_{gas}$ concentrations were consistently lower than atmospheric levels above. In addition, snowpack $Hg^0_{gas}$ declined with depth in the snowpack and were lowest in the underlying soil, showing evidence of a consistent $Hg^0_{gas}$ concentration gradient from the atmosphere to surface snow to tundra soils.

The top of the snowpack (ranging between 2 and 12 cm depth below the atmosphere depending on snow depth) generally showed highest $Hg^0_{gas}$ concentrations close to concentrations measured in the atmosphere. This pattern is inconsistent with other arctic snowpack measurements that showed atmospheric $Hg^0_{gas}$ concentrations higher than those in snowpack (Steffen et al., 2014). Indeed, the uppermost snowpack $Hg^0_{gas}$ concentrations can reach 3-times the atmospheric levels in the interior Antarctic regions (Angot et al., 2016b). It also differed to patterns observed in lower latitude snowpacks: in the Rocky

Mountains, for example, the upper snowpack showed strong enrichments of $Hg^0_{gas}$ throughout most of the winter (i.e., up to 6-times higher concentrations than in the atmosphere; Fig. 3b, Faïn et al., 2013). Such $Hg^0_{gas}$ concentration enrichments were attributed to strong photochemically initiated reduction of snow-bound $Hg^{II}$ to $Hg^0_{gas}$ (Lalonde et al., 2002). The implications of $Hg^0_{gas}$ production is that subsequent volatilization of the $Hg^0_{gas}$ from the porous snowpack to the atmosphere can alleviate atmospheric deposition loads, and it is estimated that globally 50% of snow-bound Hg is volatilized back to the atmosphere

prior to snowmelt (Corbitt et al., 2011). Our trace gas concentration measurements showed that $Hg^0_{gas}$ re-volatilization does not occur in this interior tundra snowpack during most of the winter. An absence of direct solar radiation likely explains the lack of photochemical $Hg^0_{gas}$ formation and volatilization between December through mid-January. Yet, springtime is a photochemically active period in the arctic when strong $Hg^0_{gas}$ volatilization from snow has been reported further north along the Arctic Ocean coast (Brooks et al., 2006; Kirk et al., 2006). Even in late spring, when abundant solar radiation is present

($400–600 \ W \ m^{-2}$), however, $Hg^0_{gas}$ volatilization losses were rare and largely limited to periods of active AMDEs. We speculate that a reason for the general lack of $Hg^0_{gas}$ formation and volatilization in snow includes substrate limitation due to low snow $Hg_{tot}$ concentrations (Fig. 2). An alternative possibility may be that our sampling setup (between 5 and 7 cm below the surface during the three main AMDEs) may have limited our ability to detect and observe photo-reduction processes that may occur only in the upper few cm of the snowpack surface (King and Simpson, 2001; Poulain et al., 2004). However, using

the same measurement system, $Hg^0_{gas}$ concentration enhancements in temperate snowpacks were large (up to 8 ng m$^{-3}$) and detectable up to a depth of >90 cm from the snowpack surface (Fig. 3b). Unlike in Faïn et al. (2013), we also did not observe $Hg^0_{gas}$ formation after fresh snowfall, although it also is important to note that snowfall amounts at Toolik were much lower than in temperate snowpack (Cherry et al., 2014).

     During March and April, snowpack $Hg^0_{gas}$ concentrations were highly variable (Figs. 4c and f) following $Hg^0_{gas}$ concentration

changes in the atmosphere above, indicating an apparently high snowpack diffusivity (Fig. S2). During these time periods, snowpack $Hg^0_{gas}$ concentrations in the top snowpack at times exceeded concentrations in the atmosphere above (less than 5% of the time), and these occurrences were mainly related to periods of AMDEs when $Hg^0_{gas}$ depletion occurred in the overlying atmosphere. Our measurements of $Hg^0_{gas}$ showed that early spring was the only time period when we observed small rates of $Hg^0_{gas}$ formation in the uppermost snowpack layer, suggesting some photochemical reduction and re-volatilization of $Hg^0_{gas}$

after AMDE-Hg deposition. However, $Hg^0_{gas}$ production was small, limited in time, and no photochemical $Hg^0_{gas}$ production or re-emission was observed in deeper snow layers, suggesting that the process was limited to the snowpack surface. These patterns in March and April were also consistent with flux measurements when we observed periods of net $Hg^0_{gas}$ emission from the tundra ecosystem to the atmosphere (Obrist et al., 2017), in support of the typical Hg dynamics often reported during

AMDEs ($Hg^{II}$ deposition followed by photochemical reduction and $Hg^0_{gas}$ re-emission; Ferrari et al., 2005). We propose that, in addition to relatively infrequent and generally weaker AMDE activity, rapid photochemical re-emission losses of Hg following AMDEs render these events relatively unimportant as a deposition source of Hg in this interior arctic tundra site. We provided support for this notion using stable Hg isotope analysis in soils from this site in Obrist et al. (2017), which showed that atmospheric $Hg^0_{gas}$ is the dominant Hg source to the interior tundra snowpack accounting for over 70% of Hg present.

**3.2.2 Snowpack diffusivity of trace gases**

A key question pertaining to the wintertime snowpack $Hg^0_{gas}$ concentration profiles and measured deposition is if the observed $Hg^0_{gas}$ deposition and concentration declines in the snowpack are driven by $Hg^0_{gas}$ sinks in the snowpack or by $Hg^0_{gas}$ uptake by underlying tundra soils. Sinks of $Hg^0_{gas}$ in the snowpack have been observed in a few studies (Dommergue et al., 2003; Faïn et al., 2008, 2013) and have been attributed to dark oxidation of $Hg^0_{gas}$ to divalent, non-volatile $Hg^{II}$, possibly including

oxidation by halogen species, $O_3$, or related to NOx chemistry. To address this question, we compared the ratios of $Hg^0_{gas}$ to $CO_2$ gradients in the snowpack to determine commonality or differences between sinks and sources of both gases. Because $CO_2$ in the atmosphere is relatively stable in winter and soils are the only wintertime source, $CO_2$ can be used to assess how the snowpack affects diffusion and advective exchange processes between soils and the atmosphere. Comparing $Hg^0_{gas}$ to $CO_2$ allows assessment of whether $Hg^0_{gas}$ concentrations in the snowpack are driven by processes in the underlying soils (i.e., similar

to $CO_2$) or if in-snowpack chemistry affects $Hg^0_{gas}$ concentration profiles. The gas diffusion model, based on Fick's first law of diffusion, is defined as follows, Eq. (1):

$$F = -D \left( \frac{\delta C}{\delta z} \right) \tag{1}$$

where $F$ is the molecular flux in the snowpack airspace (mol m$^{-2}$ s$^{-1}$), $D$ is the diffusivity in the snowpack airspace (m$^2$ s$^{-1}$), and $\delta C/\delta z$ is the gas concentration gradient in the snowpack integrated in the snow depth (mol m$^{-4}$).

Since diffusivity is determined by both snowpack porosity and tortuosity—both of which are poorly known and not directly measured—, we used the flux ratios between $Hg^0_{gas}$ and $CO_2$ to determine if both gases show similar flux behavior across the snowpack (Faïn et al., 2013), Eq. (2):

$$\frac{F_{Hg^0_{gas}}}{F_{CO_2}} = \frac{D_{Hg^0_{gas}}}{D_{CO_2}} \times \frac{\Delta_{Hg^0_{gas}}}{\Delta_{CO_2}} \tag{2}$$

where $\Delta_{Hg^0_{gas}}$ and $\Delta_{CO_2}$ are the $\delta C/\delta z$ gradients for both $Hg^0_{gas}$ and $CO_2$, respectively. Assuming similar gas diffusivity for

both $Hg^0_{gas}$ and $CO_2$, the ratio of concentration gradients of the two gases ($\Delta_{Hg^0_{gas}}/\Delta_{CO_2}$) gives direct information about their respective flux ratios between different snowpack trace gas inlets. Please note that these fluxes are in the opposite direction.

We focused our analysis of $Hg^0_{gas}$ and $CO_2$ concentration gradients at Toolik for the month of January 2015, when the snow height was among the highest (approximatively 40 cm), and when strong decreases in interstitial $Hg^0_{gas}$ concentrations from the top to the bottom of the snowpack were present. At this time, soils still were a relatively active source of $CO_2$ to the snowpack (Fig. 5), facilitating a comparison to the soil $CO_2$ source. In contrast to $Hg^0_{gas}$ (Fig. 5a), profiles for $CO_2$ showed

strong increases in concentrations with increasing depth in the snowpack (Fig. 5b). Highest $CO_2$ concentrations were present in the soil (up to 5000 µmol mol$^{-1}$, data not shown), and these patterns are consistent with an expected source of soils for $CO_2$ and diffusive and advective mixing of $CO_2$ produced in snow through the snowpack with the atmosphere (Liptzin et al., 2009; Oechel et al., 1997). Analysis of $\Delta_{Hg^0_{gas}}/\Delta_{CO_2}$ ratios showed no statistically significant differences from the top to the bottom of the snowpack, as evidenced from calculated gradients between 0 to 10 cm, 10 to 20 cm, and 20 to 30 cm heights (Fig. 5c).

The constant and negative ratios between $CO_2$ and $Hg^0_{gas}$ and the fact that $CO_2$ is largely non-reactive in snowpack hence indicates that $Hg^0_{gas}$ also was not subject to snowpack chemical reactions; both profiles are affected by underlying soil processes, i.e., soil sources for $CO_2$ and soil sinks (for $Hg^0_{gas}$). These wintertime atmosphere–snowpack–soil $Hg^0_{gas}$ concentration profiles at Toolik were also consistent with a measured net deposition of $Hg^0_{gas}$ throughout winter using flux measurements (Figs. 2 and 4; Obrist et al., 2017). Both net flux measurements, combined with snowpack $Hg^0_{gas}$ concentration

profiles, hence suggest that a soil $Hg^0_{gas}$ sink was active throughout the Arctic winter, notably under very cold wintertime soil temperatures as low as −15 °C. Such soil $Hg^0_{gas}$ sinks were previously reported to occur in temperate soils (Obrist et al., 2014), although the mechanisms for the $Hg^0_{gas}$ sinks are currently not clear. It is notable that $\Delta_{Hg^0_{gas}}/\Delta_{CO_2}$ ratios in the upper snowpack (i.e., between 20 and 30 cm height) were more variable compared to lower snowpack heights, which we attribute to much smaller concentrations differences for both $CO_2$ and $Hg^0_{gas}$ between these inlets.

**3.3 Snowbound mercury in the interior arctic snowpack**

**3.3.1 Spatial patterns**

Snow samples were analyzed at Toolik for $Hg_{tot}$ and $Hg_{diss}$ (Fig. 2 and Table S1). Concentrations in snowpack collected over the tundra averaged $0.70 \pm 0.98$ ng L$^{-1}$ for $Hg_{tot}$ concentrations and $0.17 \pm 0.10$ ng L$^{-1}$ for $Hg_{diss}$ concentrations (both seasons, average of entire snowpack height). Total Hg concentrations were always higher than $Hg_{diss}$ levels, likely due to impurities and deposition of Hg associated with plant detritus or soil dust, and showed higher variability in $Hg_{tot}$ concentrations compared

to $Hg_{diss}$. We thus focused our discussions on $Hg_{diss}$ data. Measurements performed at Toolik showed very low levels compared to many other high latitude studies, with $Hg_{diss}$ concentrations averaging $0.17$ ng L$^{-1}$, and ranging between 0.08 and 1.15 ng L$^{-1}$. This is generally lower than Hg concentrations in interior Arctic sites reported by Douglas and Sturm (2004) (i.e., $Hg_{diss}$ concentrations between 0.5 and 1.7 ng L$^{-1}$) and at the low end of concentrations found in Artic studies along the coastal

zone (0.14–820 ng L$^{-1}$, for both $Hg_{diss}$ and $Hg_{tot}$; Douglas et al., 2005; Douglas and Sturm, 2004; Ferrari et al., 2004, 2005; Kirk et al., 2006; Nerentorp Mastromonaco et al., 2016; St. Louis et al., 2005; Steffen et al., 2002). The low concentrations we measured result in very small pool sizes of $Hg_{diss}$ stored in the snowpack during wintertime compared to temperate studies

(Pearson et al., 2015). At Toolik, snowpack pool sizes amounted to 26.9 and 19.7 ng m$^{-2}$ during peak snowpack and prior to the onset of snowmelt in 2014–2015 and 2015–2016, respectively.

The snowpack sampled over the adjacent frozen lake showed Hg$_{tot}$ and Hg$_{diss}$ concentrations of 0.80 ± 0.61 and 0.15 ± 0.08 ng L$^{-1}$, respectively (Table S1). These values were not statistically different from concentrations measured in the tundra snowpack. Snowpack Hg$_{diss}$ loads on the frozen lake were lower (6.2 ± 0.2 ng m$^{-2}$), i.e., only about ¼, compared to snowpack Hg$_{diss}$ load on the adjacent tundra (23.3 ± 5.0 ng m$^{-2}$). Three reasons may explain the large difference between lake and tundra snowpack Hg loads: (1) the lake did not accumulate the snowpack on open water prior to the lake surface freezing in the early fall (Sturm and Liston, 2003); (2) low surface roughness over the lake likely prevent settling of snowfall and facilitate remobilization of snow by wind transport (Essery et al., 1999; Essery and Pomeroy, 2004); and (3) the lake ice is warmer than the tundra soil resulting in higher sublimation over the lake. The implications of the latter process is a reduction of direct atmospheric deposition over Arctic lakes, and is consistent with studies that estimated that annual Hg contribution to Arctic lakes via direct wet deposition is small, generally less than 20% of total deposition (Fitzgerald et al., 2005, 2014). Spatial redistribution of snow across the tundra landscape further implies that both wet deposition and snow accumulation rates are variable, leading to spatial heterogeneity of snowmelt Hg inputs.

Most Arctic studies of snowpack Hg have been performed close to the coast (i.e., Alert and Barrow), and few studies include inland sites such as Toolik (Douglas and Sturm, 2004). In our study, measurements of Hg$_{tot}$ and Hg$_{diss}$ in the snowpack across a large North slope transect (about 170 km from Toolik to the Arctic Coast) in March, 2016 showed concentrations of 0.70 ± 0.79 and 0.24 ± 0.20 ng L$^{-1}$, respectively (Fig. 6 and Table S3). Concentrations in Hg$_{diss}$ of the five northernmost locations (<100 km distance from the Arctic Ocean) were statistically significantly ($p < 0.05$, Kruskal-Wallis test) higher compared to those measured at the four stations located in the interior tundra (>100 km), which included the Toolik site where mean Hg$_{diss}$ concentrations were 0.33 ± 0.22 and 0.11 ± 0.07 ng L$^{-1}$ for the same period, respectively. These patterns are consistent with previous observations in Alaska in springtime that suggested an ocean influence leading to higher Hg deposition, possibly linked to the presence of halogens (Douglas and Sturm, 2004; Landers et al., 1995; Snyder-Conn et al., 1997). We propose that low snowpack Hg concentrations (<0.5 ng L$^{-1}$ for Hg$_{diss}$) are common in inland northern Alaska areas, and that the interior arctic snowpacks exhibit lower levels compared to coastal locations that are subjected to more significant ocean influences and impacts by AMDEs.

### 3.3.2 Seasonal patterns

Surface snow that was collected throughout the season can serve as an estimate for atmospheric wet deposition Hg concentrations and loads (Faïn et al., 2011). Concentrations of Hg$_{tot}$ and Hg$_{diss}$ in the surface snow layer (top 3 cm only) averaged 0.53 ± 0.39 ng L$^{-1}$ and 0.26 ± 0.26 ng L$^{-1}$, respectively (Fig. 7 and Table S1), which were not statistically significantly different compared to that of full snow pits or bottom snow layers. Both, low concentrations measured in surface snow, as well as low pool sizes as discussed above, suggest low wet deposition rates during winter at our inland arctic sites. However, estimation of deposition loads using snow collection can be compromised by quick re-volatilization losses of Hg

from fresh snowfall (within the first few hours, e.g., Faïn et al., 2013), or snowmelt losses, but we do not consider these processes to be important at this site. The low $Hg_{diss}$ concentrations measured in surface snow ($0.26 \pm 0.26$ ng L$^{-1}$) are lower than the 10$^{th}$ percentile of wet deposition Hg concentrations reported for Kodiak Island in Alaska during the same time period (National Atmospheric Deposition Program, 2017). Also, snowfall $Hg_{diss}$ concentrations measured at Alert were between 100 and 200-times higher than in our measurements (A. Steffen, personal communication). Using median concentrations in the surface snow multiplied by the amount of wet deposition for each snow-covered season, we estimated the $Hg_{diss}$ load annually deposited by snowfall to 41.3 and 15.3 ng m$^{-2}$ in the 2014–2015 and 2015–2016 winters, respectively. This is 1/100 of values recently provided from a coastal location 400 km northwest of our study site (Douglas et al., 2017) and 1/200 of long-term measurements from Alert between 1998 and 2010 (A. Steffen, personal communication).

Little temporal variation in snowpack Hg concentrations was observed between the early season snowpack evolving mainly under darkness and the late-season snowpack exposed to solar radiation (Figs. 2 and 6), although some temporal differences were evident during March and April when AMDEs were present in the region. Snowpack $Hg_{diss}$ concentrations averaged 0.16 ng L$^{-1}$ both during the completely dark period (i.e., December and January) and after March 1$^{st}$. Such patterns support measurements of $Hg^0_{gas}$ throughout the winter that indicated the snowpack to be a relatively inert matrix with little redox processes affecting Hg concentrations (oxidation of $Hg^0_{gas}$ or reduction of $Hg^{II}$). An apparent trend in surface snow, however, emerged during springtime, when both $Hg_{tot}$ and $Hg_{diss}$ concentrations exceeded 1 ng L$^{-1}$ (i.e., 4-times the average values observed through the rest of the season; Fig. 7). This was a period when AMDEs occurred at this site, as evident by depletions of atmospheric $Hg^0_{gas}$ with formation and deposition of oxidized atmospheric $Hg^{II}$ (Obrist et al., 2017; Van Dam et al., 2013). Surface snow Hg concentration enhancements during AMDEs are commonly reported in polar regions, with at times Hg concentration enhancements up to 100-times the base concentration in the Arctic (Lalonde et al., 2002; Lindberg et al., 1998; Poulain et al., 2004; Steffen et al., 2002). The presence of AMDEs generally results in increased deposition of Hg to snow and ice surfaces, yet such additional deposition often is short-lived due to the photochemical re-emission of $Hg^0_{gas}$ (Kirk et al., 2006). In our study, we did not have sufficient temporal resolution of snow sampling during the period of AMDEs to closely track the fate of Hg deposition during AMDEs and subsequent re-emissions. However, we find that snow Hg enhancements during AMDEs were much lower than at coastal sites (e.g., Steffen et al., 2014), but a coarse temporal sampling could just have missed peak snow Hg levels at this site. We also found that after AMDEs, snow $Hg_{diss}$ in surface snow declined to levels as was observed prior to AMDEs, and no concentration enhancements were observed deeper in the snowpack. This is consistent with observations of net $Hg^0_{gas}$ volatilization during that time. The fact that we found no lasting impact of AMDEs on snow Hg concentrations, which also were supported by stable Hg isotope analysis (Obrist et al., 2017), may be due to the large distance to the coast from our study site and the scarcity of AMDEs—and O$_3$ depletion events—that occur at this inland arctic location (Van Dam et al., 2013).

Concentrations of $Hg_{diss}$ measured in the snowpacks at Toolik did not show consistent vertical patterns (Fig. 2). Indeed, the upper snowpack $Hg_{diss}$ concentrations were not significantly different from those in the deeper layers, which is in contrast to patterns observed in arctic snowpacks (Ferrari et al., 2004), as well as in alpine ones (Faïn et al., 2011), where strong

concentration enhancements (i.e., more than 2-times the average snowpack concentrations) were observed in the top 3 cm of the snowpack. Seasonal measurements at Toolik indicate a generic lack of atmospheric gaseous $Hg^{II}$ during most of the year and very low amounts of total $Hg^{II}$ deposition, i.e., wet, aerosols, plus gaseous $Hg^{II}$ (Obrist et al., 2017). The lack of significant $Hg^{II}$ dry deposition would prevent a Hg enhancement in surface snow, and also is consistent with the low pool sizes of Hg in this tundra snowpack. Further support of this notion also includes that snow collected at the surface throughout the arctic winter and spring was not statistically different from snow Hg concentrations contained in the entire snowpack ($0.26 \pm 0.26$ vs $0.17 \pm 0.10$ ng $L^{-1}$, respectively). Yet, another factor to explain a lack of depth gradients in snow Hg concentrations may include that snow layers can be continuously mixed and redistributed by wind gust (e.g., wind speed of Toolik were >5 m $s^{-1}$ 12% of the time) across the landscape in the Arctic (Cherry et al., 2014).

### 3.4 Origin of mercury in the interior arctic snowpack

#### 3.4.1 Cation and anion concentrations

Major cations ($Ca^{2+}$, $K^+$, $Mg^{2+}$, $Na^+$, and $NH_4^+$) and anions ($Cl^-$, $NO_3^-$, and $SO_4^{2-}$) were measured in snowpack and surface snow samples at Toolik to assess the chemical composition and potential origins for Hg in the snowpack (Table 1). Concentrations of these compounds were comparable to other inland Alaskan sites and, similar to concentrations of Hg, were lower than data reported from several arctic coastal locations (de Caritat et al., 2005; Douglas and Sturm, 2004). Surface snow samples (top 3 cm) generally showed somewhat higher $Cl^-$ and $Na^+$ concentrations and lower $Mg^{2+}$ and $K^+$ concentrations than samples collected across the entire snowpack height, although only $Mg^{2+}$ and $Na^+$ were significantly different ($p < 0.005$ and $p < 0.05$, respectively). Comparison between tundra and lake snowpack locations showed no statistical differences in elemental concentrations.

Spearman correlation coefficient ($\rho$) between $Hg_{diss}$ and major ion concentrations were calculated for tundra and lake snowpack samples and surface snow collected over the tundra (Table 2). Using a correlation matrix, three groups of correlated major ions could be determined in the snowpack over the tundra: (1) $NH_4^+$ and $SO_4^{2-}$; (2) $Ca^{2+}$, $Mg^{2+}$, and $NO_3^-$; (3) $Cl^-$, $K^+$, and $Na^+$. In the tundra snowpack, $Hg_{diss}$ was not statistically significantly ($-0.22 < \rho < 0.11$) correlated to any of these major ion groups when considering the entire depth of the tundra snowpack. Relationships, however, were present in surface snow over the tundra where $Hg_{diss}$ was correlated ($\rho$ up to 0.80) with $Ca^{2+}$, $Cl^-$, and $K^+$, indicating that $Hg_{diss}$ may have originated from a mix of natural sources possibly linked to both mineral dust ($Ca^{2+}$) and sea spray ($Cl^-$). The lack of strong correlation between $Hg_{diss}$ and $Na^+$ ($\rho = 0.30$) in surface snow samples may indicate that a part of $Cl^-$ originated from mineral dust as $CaCl_2$. A minor influence of sea salt was consistent with coastal observations that showed the highest Hg concentrations close to the Arctic Ocean related particularly to active bromine chemistry (Fig. 6; Douglas and Sturm, 2004). In addition, local or regional dust from rock and soil weathering contributed to the wintertime Hg deposition, particularly at interior sites close to the Brooks Range where higher snow pH reported were from mineral dust that contained carbonates (Douglas and Sturm, 2004). Indeed, the mountain influence was dominant during the two snow-covered seasons at Toolik where 50% of snow events and 80% of

dry periods (i.e., periods without snowfall, 90% of the time) came from the south (i.e., Brooks Range). An additional group of correlated elements was identified in surface snow samples over the tundra: $NH_4^+$, $NO_3^-$, and $SO_4^{2-}$. Note that the low number of lake snowpack samples ($\leq 12$) did not allow us to perform a meaningful correlation matrix analyses for lake snowpack samples.

To further visualize the relationships between analytes, we plotted a ternary diagram using three end-members according to Garbarino et al. (2002), Krnavek et al. (2012), Poulain et al. (2004), and Toom-Sauntry and Barrie (2002) (Fig. 8). We considered $Ca^{2+}$ as one end-member to represent a potential crustal signature, a second end-member with $Cl^-$ as a sea salt signature, and a third end-member with $SO_4^{2-}$ as a potential anthropogenic signature, i.e., from regional or long-range transport. Since sea salt $SO_4^{2-}$ represented on average less than 1.2% of total $SO_4^{2-}$ according to the calculation of Norman et al. (1999),

we consider $SO_4^{2-}$ not indicative of an ocean source. The different snow types (surface snow over the tundra, tundra snowpack, and lake snowpack) are presented with different colors in Fig. 8, and $Hg_{diss}$ concentrations are represented by different symbol sizes. Relative concentrations of $Cl^-$ (i.e., sea salt influence) showed statistically significant differences between snow samples collected over the tundra and those collected over the frozen lake (on average, 14 and 24% of proportion based on normality data, respectively; $p < 0.05$). However, no statistically significant differences were observed for relative concentrations of $Ca^{2+}$

and $SO_4^{2-}$ between tundra and lake locations. In general, snow surface samples showed low $SO_4^{2-}$ and $Cl^-$ relative concentrations ($<30\%$) compared to integrated snowpack samples. Overall, $Hg_{diss}$ concentrations were weakly correlated, except according to the $SO_4^{2-}$ relative concentrations: $Hg_{diss}$ concentrations averaged 0.10 and 0.17 ng $L^{-1}$ for $>30\%$ and $<30\%$ of $SO_4^{2-}$, respectively ($p < 0.005$). These patterns indicate that anthropogenic influences from combustion processes were minor or absent for snow Hg deposition. In fact, Alaska generally showed the lowest $SO_4^{2-}$ concentrations among arctic sites

(de Caritat et al., 2005). Norman et al. (1999) also reported relatively small contributions of anthropogenic $SO_4^{2-}$ in snow at Alert (Canada). From this, we propose that the Hg sources in the arctic snowpack is mainly derived from local lithological erosion, and that Arctic Ocean sources are minor contributions. However, this is not likely the case of $Hg^0_{gas}$ in tundra soils which mainly derived from global sources (Obrist et al., 2017). It should be noted that the proximity of Toolik with a busy road in the Arctic (the Dalton Highway) may influence our measurements, but this is difficult to evaluate.

The lack of consistent statistically significant associations between major ions and $Hg_{diss}$ across the entire snowpack depth (Table 2a) further suggest that initial snowfall Hg content was maintained and largely unaltered after deposition, with no clear accumulation or depletion zones as found in other snowpacks (Ferrari et al., 2005; Poulain et al., 2004; Steffen et al., 2014). We found a small relative enrichment of alkaline earth elements in snowpack samples compared to surface snow, which indicates some additional contributions of local mineral dust, yet this did not result in a measurable increase in snowpack Hg

levels. Hence, we suggest no significant additional deposition of Hg (e.g., by dry deposition of gaseous or particulate Hg) to exposed older snow consistent with the lack of correlation to pollution tracers ($SO_4^{2-}$ and $NO_3^-$). We also suggest an absence or minor importance of re-emission losses or elution losses from snow melt as occurs in temperate snowpacks (discussed in Faïn et al. (2013) and Pearson et al. (2015)). Elution losses are unlikely, given that no temperatures above freezing were present

in the Arctic until May, and atmospheric re-emissions losses of volatile $Hg^0_{gas}$ were not important in this arctic snowpack for most of the season as discussed above.

### 3.4.2 Stable oxygen and hydrogen isotope signatures

Oxygen ($^{18}O$) and hydrogen ($^2H$) isotopes are frequently used as tracers for precipitation sources (Gat, 2010). The stable isotope signatures in surface snow samples collected at Toolik are presented in a $\delta^2H$ vs $\delta^{18}O$ diagram for different ranges of $Hg_{diss}$ concentrations and different sampling dates (Fig. 9a). All the samples were distributed close to the global meteoritic water line (Craig, 1961). Despite a large variability in values (from −18.3 to −41.3‰ for $\delta^{18}O$ and from −140 to −314‰ for $\delta^2H$), samples collected on the same date were relatively close (mean standard deviation of 0.88 and 6.5‰, respectively). No clear relationships were observed between isotope signatures and $Hg_{diss}$ concentrations (with size scale in Fig. 9) across the entire spectrum of values. However, samples with high $Hg_{diss}$ concentrations (e.g., the three highest measured in April 2nd, 2016) and low $Hg_{diss}$ concentrations (e.g., samples below the detection limit in December 5th, 2015) were found clustered together at similar $\delta^{18}O$ and $\delta^2H$ values. The $\delta^{18}O$ values were also plotted against air temperatures ($T_{air}$) during the snowfall events (Fig. 9b). A statistically significant linear relationship was found between the two variables ($r^2 = 0.50$) with the lowest $\delta^{18}O$ values being measured during the coldest temperatures. Neither the origin of precipitation as shown by the wide range of stable isotope ratios, nor the physical conditions that often cause isotopic variation in precipitation (e.g., air temperatures that explain up to 50% of isotopic values via mass effects; Siegenthaler and Oeschger, 1980), shaped the Hg concentrations measured in the snowpack.

### 4 Conclusions

In this study, we investigated snow Hg dynamics in the interior arctic tundra at Toolik Field Station, Alaska, simultaneously analyzing Hg in: (1) the gas-phase ($Hg^0_{gas}$) of the atmosphere, interstitial snowpack, and soil pores; and (2) in the solid phase in snow ($Hg_{tot}$ and $Hg_{diss}$). Gaseous $Hg^0$ in the atmosphere–snowpack–soil continuum showed consistent concentration patterns throughout most of the snow season with the arctic tundra soil serving as a continuous sink for $Hg^0_{gas}$, important to consider in Arctic Hg cycling. To our surprise, photochemical formation of $Hg^0_{gas}$ in the snowpack was largely absent and played a minor role in the interior tundra largely limited to periods of active AMDEs. These observations are in contrast with strong photochemical formation of $Hg^0_{gas}$ in surface snow observed at temperate sites and along the arctic coast, resulting in significant photochemical losses of $Hg^0_{gas}$ from these snowpacks. This calls for a regional adjustment of photochemical $Hg^0_{gas}$ losses from the snowpack in models, which should have different treatment for the arctic snowpack compared to temperate snowpacks. Small $Hg_{diss}$ enhancements were temporarily observed in surface snow during springtime, when AMDEs were present, reflecting the typical sequence of Hg deposition to the top snowpack followed by fast photochemical volatilization losses of $Hg^0_{gas}$ during that time. At this interior arctic site, AMDEs, however, resulted in negligible deposition loads. Low concentrations of both $Hg_{tot}$ and $Hg_{diss}$ were measured in the snowpack across this northern Alaska region, resulting in a small

reservoir of Hg stored in this snowpack available for potential mobilization during snowmelt (<30 ng m$^{-2}$ for Hg$_{diss}$). These low values suggest that wet Hg deposition via snow is not a major source of Hg to this interior arctic site, a notion we previously supported by direct measurements and stable Hg isotopes that showed that two thirds of the Hg source are derived from Hg$^0_{gas}$ deposition. Multielement analysis of surface snow (top 3 cm) indicated that arctic snowpack Hg originated from a mix of diffuse and likely natural sources, including local mineral dust (associated with Ca$^{2+}$ and Mg$^{2+}$) and, to a lesser extent, regional marine sea spray (associated with Cl$^-$ and Na$^+$).

## Acknowledgements

We thank Toolik Field Station staff for their support in this project over two years, especially Jeb Timm, Joe Franish, and Faye Ethridge, for helping with snow collection. We also thank Martin Jiskra (Geosciences Environnement Toulouse) and Christine Olson (DRI) for their field support, Christopher Pearson, Olivia Dillon, and Jacob Hoberg (DRI) for their support with laboratory analyses, and Dominique Colegrove and Tim Molnar (University of Colorado) for helping with field work and data processing. We finally thank Alexandra Steffen for providing mercury snow data from Alert. Funding was provided by the U.S. National Science Foundation (NSF) under award (#PLR 1304305) and cooperative agreement from National Aeronautics and Space Administration (NASA EPSCoR NNX14AN24A).

## References

La mise à jour automatique des citations est désactivée. Pour voir la bibliographie, cliquez sur Actualiser dans l'onglet Zotero.

**Table 1: Mean concentration (µg L$^{-1}$), including standard deviation (italics), of cations and anions in tundra and lake snowpack and in surface snow at Toolik Field Station.**

| location | | $Mg^{2+}$ | $Ca^{2+}$ | $Na^+$ | $K^+$ | $Cl^-$ | $NH_4^+$ | $NO_3^-$ | $SO_4^{2-}$ |
|---|---|---|---|---|---|---|---|---|---|
| tundra | surface | 7.2 | 453.0 | 112.6 | 29.4 | 228.6 | 11.3 | 265.0 | 191.3 |
| | | *6.8* | *530.8* | *104.6* | *46.7* | *232.9* | *3.9* | *187.5* | *130.6* |
| | snowpack | 32.1 | 523.5 | 58.5 | 60.8 | 137.5 | 13.2 | 202.8 | 234.0 |
| | | *34.7* | *452.1* | *38.9* | *102.3* | *113.1* | *5.4* | *104.5* | *131.8* |
| lake | | 27.8 | 784.1 | 119.1 | 23.1 | 117.5 | 12.8 | 270.5 | 181.2 |
| | | *21.8* | *403.7* | *135.9* | *28.6* | *73.4* | *2.9* | *94.0* | *78.1* |

**Table 2: Spearman's coefficient correlations (ρ, in bold if ≥0.5 or ≤−0.5) between chemical elements (dissolved Hg [Hg_diss] and major ions) in the tundra snowpack (a) and surface snow over the tundra (b).**

**a.** Tundra snowpack

|  | $Hg_{diss}$ | $Mg^{2+}$ | $Ca^{2+}$ | $Na^+$ | $K^+$ | $Cl^-$ | $NH_4^+$ | $NO_3^-$ |
|---|---|---|---|---|---|---|---|---|
| $SO_4^{2-}$ | −0.16 | 0.42 | 0.32 | 0.39 | 0.48 | 0.47 | **0.58** | 0.17 |
| $NO_3^-$ | 0.07 | **0.74** | **0.83** | **0.55** | 0.33 | **0.59** | 0.03 | |
| $NH_4^+$ | −0.22 | −0.04 | 0.03 | 0.15 | 0.35 | 0.30 | | |
| $Cl^-$ | −0.11 | 0.41 | 0.39 | **0.89** | **0.72** | | | |
| $K^+$ | −0.10 | 0.34 | 0.33 | **0.70** | | | | |
| $Na^+$ | 0.11 | 0.47 | 0.38 | | | | | |
| $Ca^{2+}$ | −0.07 | **0.90** | | | | | | |
| $Mg^{2+}$ | 0.06 | | | | | | | |

**b.** Surface snow

|  | $Hg_{diss}$ | $Mg^{2+}$ | $Ca^{2+}$ | $Na^+$ | $K^+$ | $Cl^-$ | $NH_4^+$ | $NO_3^-$ |
|---|---|---|---|---|---|---|---|---|
| $SO_4^{2-}$ | −0.08 | **0.54** | 0.14 | 0.16 | −0.08 | −0.04 | **0.74** | **0.74** |
| $NO_3^-$ | 0.14 | **0.62** | 0.28 | 0.08 | 0.07 | −0.20 | **0.57** | |
| $NH_4^+$ | −0.02 | 0.45 | 0.24 | 0.18 | −0.08 | −0.04 | | |
| $Cl^-$ | **0.63** | 0.35 | **0.69** | **0.82** | **0.86** | | | |
| $K^+$ | **0.62** | 0.45 | **0.80** | **0.78** | | | | |
| $Na^+$ | 0.30 | **0.68** | **0.56** | | | | | |
| $Ca^{2+}$ | **0.80** | 0.39 | | | | | | |
| $Mg^{2+}$ | 0.08 | | | | | | | |

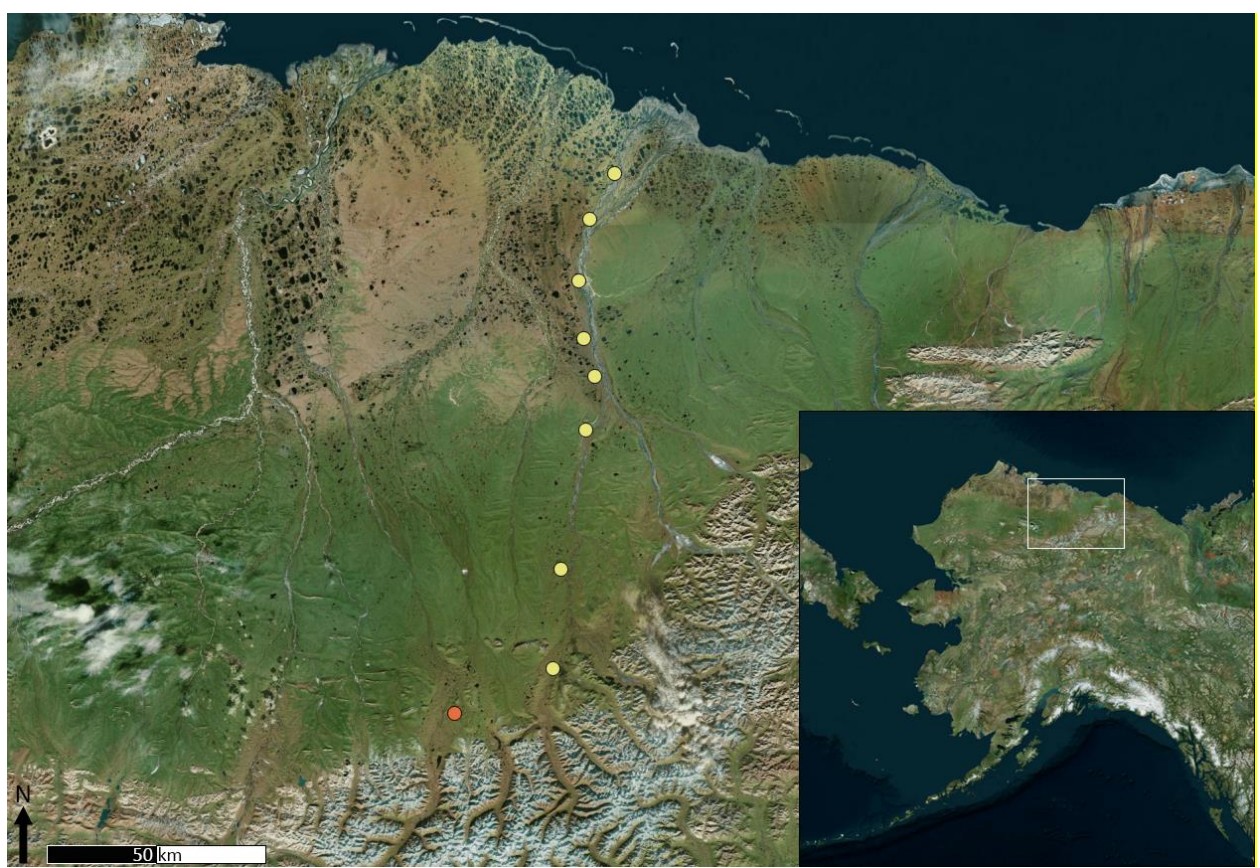

**Figure 1: Study area in northern Alaska, including Toolik Field Station (orange bullet point) and the eight transect sites (yellow bullet points). Satellite images are true color images (Earthstar Geographics SIO, 2017).**

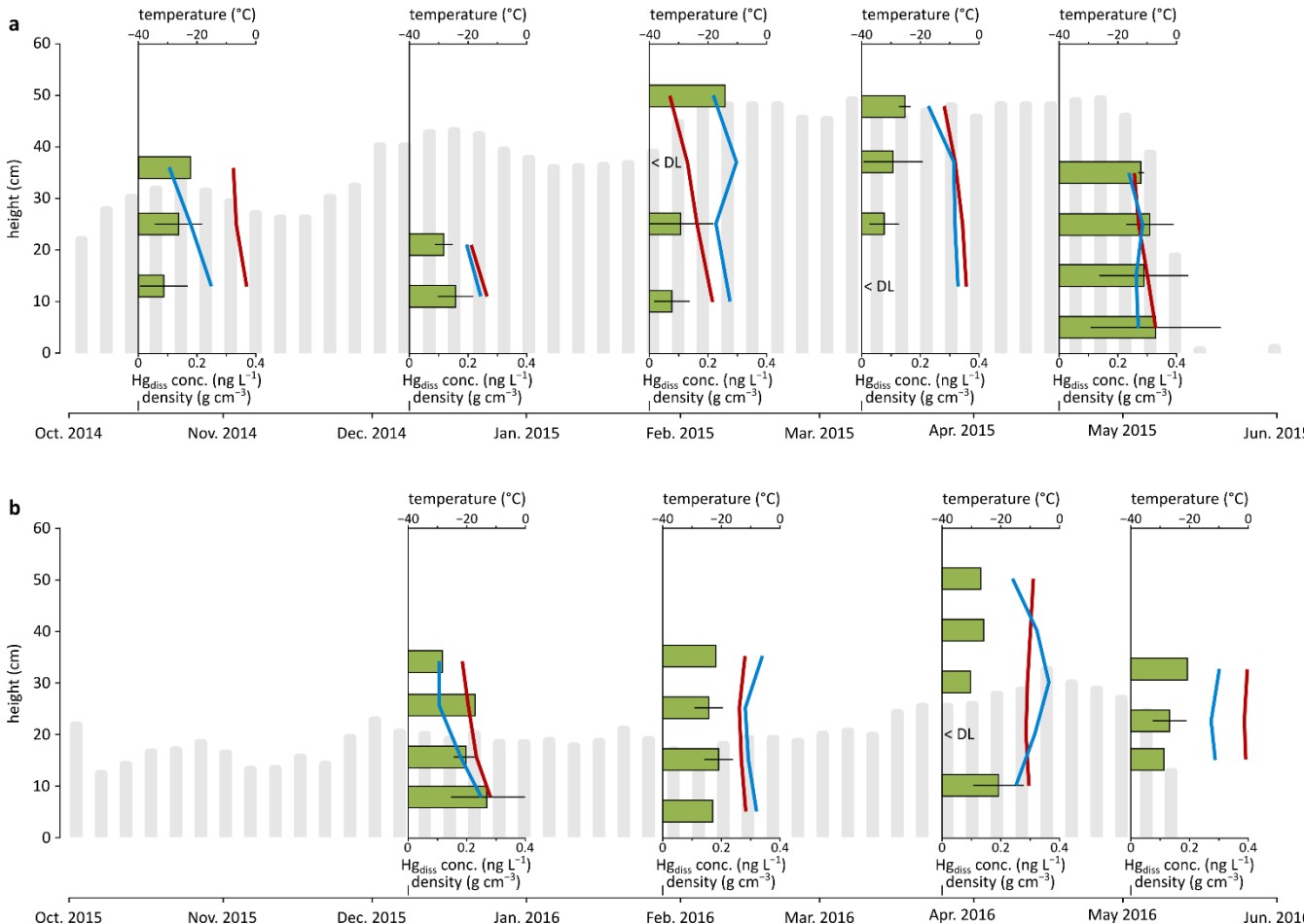

**Figure 2:** Snowpack temperatures (red lines) and densities (blue lines) and dissolved Hg concentrations (green bars, including mean values and standard deviations) for five snow pits in the 2014–2015 season (a) and four snow pits in the 2015–2016 season (b) over the Arctic tundra at Toolik Field Station. The gray bars illustrate the average snow heights.

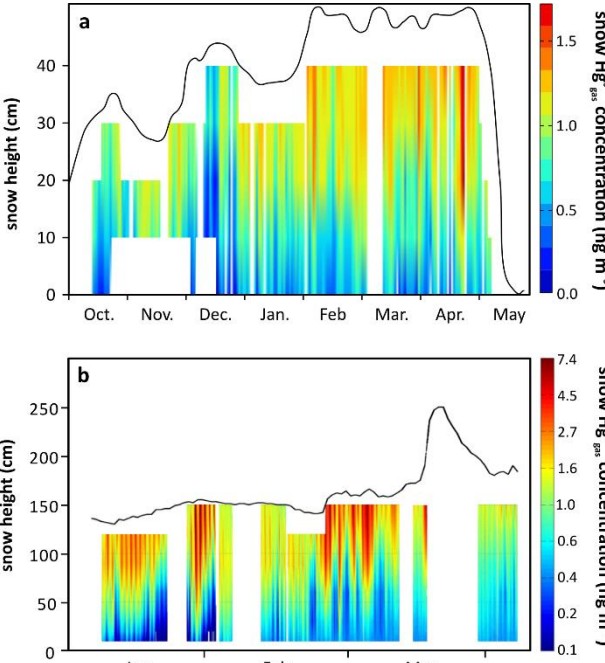

**Figure 3: Gaseous Hg$^0$ concentration profiles in snowpack interstitial air during the snow-covered season from October 2014 to May 2015 over the Arctic tundra measured at Toolik Field Station based on continuous observations at up to five heights in the snowpack each hour, and interpolation of this data across the entire snowpack (a). For comparison, interpolated Hg$^0_{gas}$ concentration profiles in snowpack interstitial air during the snow-covered season based on similar measurements at Niwot Ridge, Rocky Mountains, Colorado, USA, during the winter of 2009 (b) (adapted with permission; Faïn et al., 2013).**

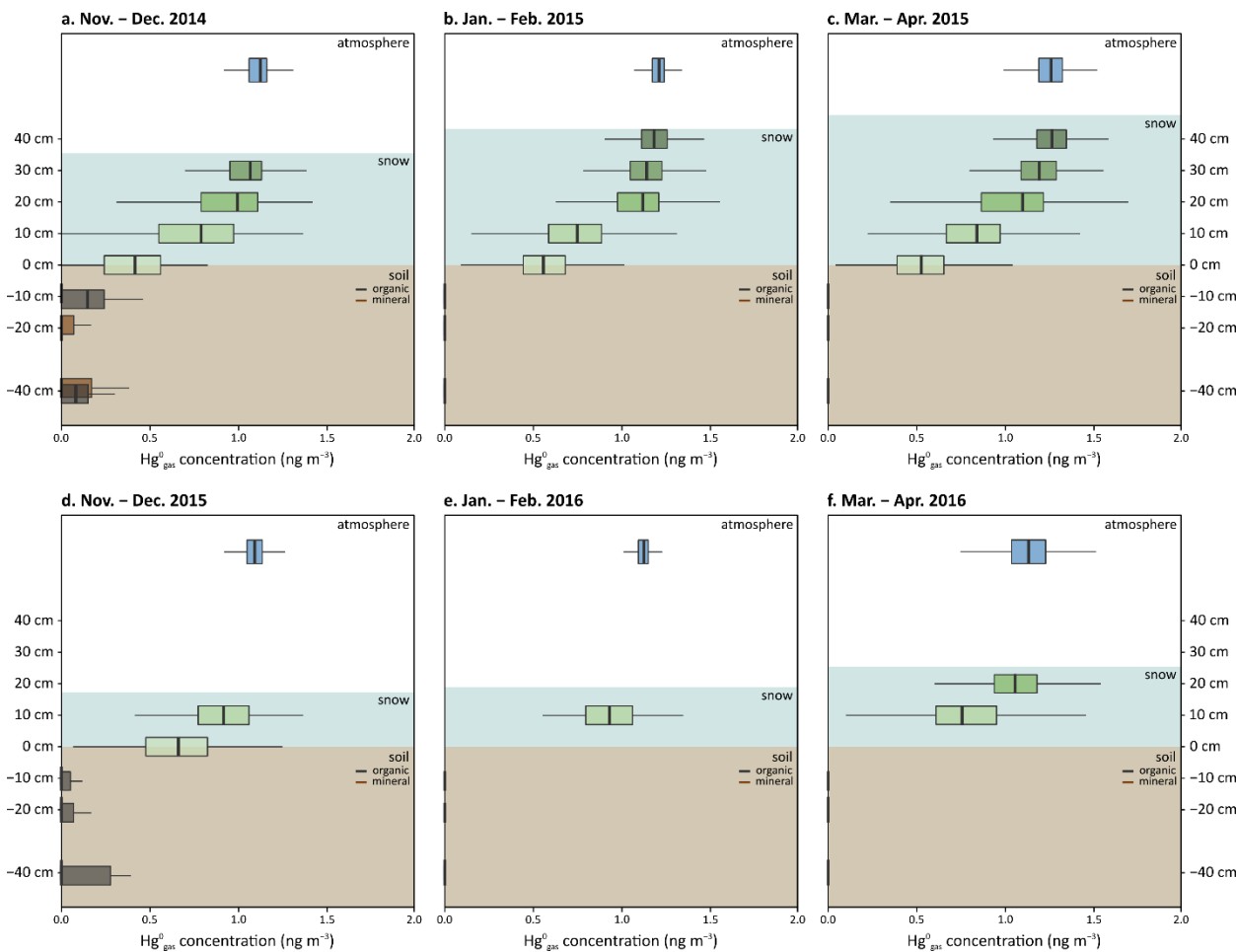

**Figure 4: Hg⁰gas concentration profiles in the atmosphere, snowpack interstitial air, and soil interstitial air in early winter (from November to December; a and d), in winter (from January to February; b and e), and in early spring (from March to April; c and f) for 2014–2015 (top panels) and 2015–2016 (bottom panels) snow-covered periods over the arctic tundra measured at Toolik Field Station.**

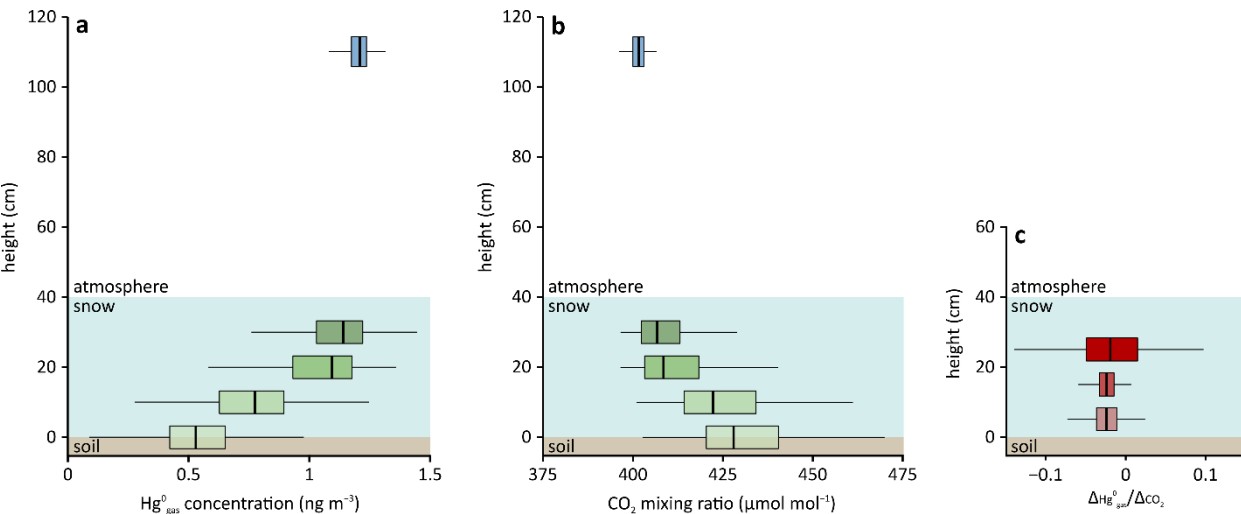

**Figure 5: Snow concentration profiles for Hg$^0_{gas}$ (a) and CO$_2$ (b) concentrations, and $\Delta_{Hg^0_{gas}}/\Delta_{CO_2}$ ratios for 0 to 10 cm, 10 to 20 cm, and 20 to 30 cm snowpack height based on daily averages (c) in January 2015 (snow height averaged 40 cm) over the arctic tundra measured at Toolik Field Station.**

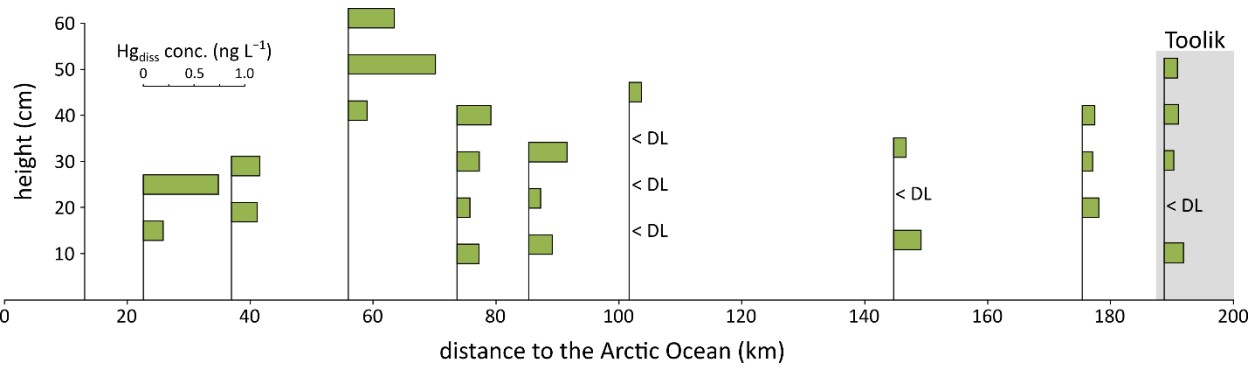

**Figure 6: Spatial pattern of dissolved Hg concentrations (Hg$_{diss}$) in snowpack profiles across the North slope transect on March 27th–28th, 2016, and comparison with Toolik Field Station (gray box) in March 25th, 2016.**

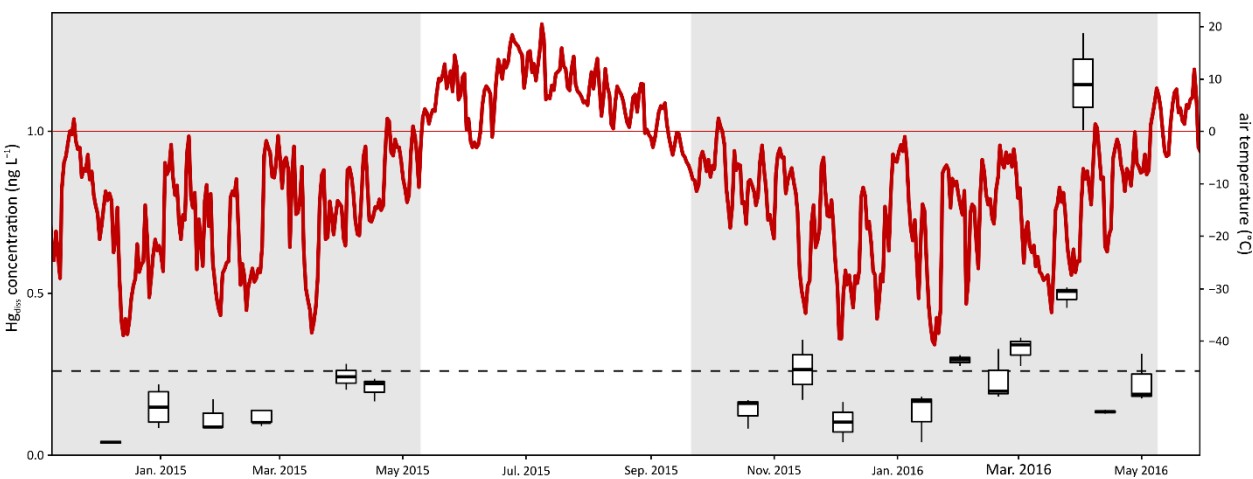

**Figure 7: Temporal pattern of dissolved Hg (Hg$_{diss}$) concentrations in surface snow samples (top 3 cm) throughout the 2014–2015 and 2015–2016 snow-covered seasons (in grey) at Toolik Field Station. The broken line indicates the average surface snow Hg$_{diss}$ concentration (0.26 ng L$^{-1}$). The red line indicates the daily average air temperature.**

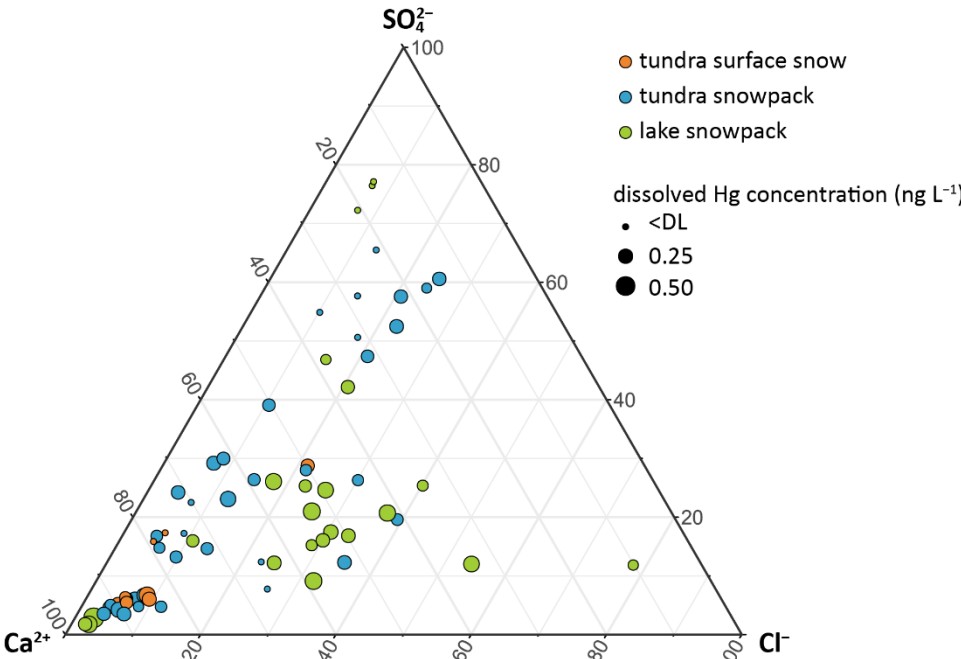

**Figure 8: Ternary diagram of tundra surface snow (orange), tundra snowpack (blue), and lake snowpack (green) samples from Toolik Field Station ordered by dissolved Hg concentration between $Ca^{2+}$, $Cl^-$, and $SO_4^{2-}$ (proportions based on meq $L^{-1}$).**

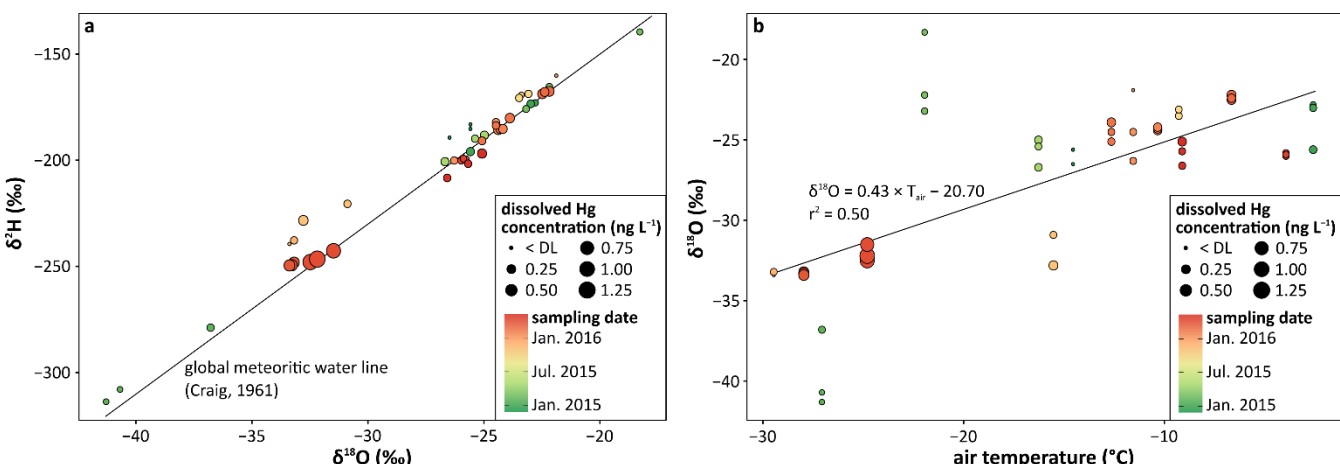

**Figure 9: Dissolved Hg concentrations in surface snow samples for 2014 to 2016 in: (a) the $\delta^2$H vs $\delta^{18}$O diagram and (b) a plot of $\delta^{18}$O vs air temperature ($T_{air}$) during the previous snowfall at Toolik Field Station.**