# Peer review of "Mercury in the Arctic tundra snowpack: temporal and spatial concentration patterns and trace-gas exchanges"

_The Cryosphere, 2017_

## Referee Comment (RC1) · Anonymous Referee #1 · 5 Jan 2018

**General comments**

The authors investigated snow Hg dynamics in the interior arctic tundra at Toolik, Alaska. They compared their results to a temperate snowpack in the Rocky Mountain, Colorado and conclude that photochemical Hg(0) losses from the snowpack in models should be adjusted and treated differently in arctic and temperate snowpacks. I think that this research provides some very interesting results and the manuscript is fairly well written. However, to my opinion, it lacks a thorough comparison to other studies performed in polar regions and a discussion on instrumental limitations. I think that the manuscript will be appropriate for publication in The Cryosphere after the authors

address the major comments discussed below.

**Major comments**

The conclusion of the manuscript (see above) derives from two main observations. Firstly, low concentrations of Hgtot and Hgdiss were measured in surface snow. Secondly, the photochemical formation of Hg(0)gas in the snowpack was largely absent.

1. I am quite surprised by the very low Hgtot concentrations in surface snow samples reported in this study, especially during depletion events. This raises concern about the potential influence of drifting snow. While the authors admit that the Toolik snowpack is subject to significant drifts and changes in snowpack height, there is no discussion on potential consequences and uncertainties.

2. While I agree that the photochemical formation of Hg(0)gas in the snowpack is most likely low compared to a temperate snowpack, I would like to see a comparison to other studies performed in polar regions. To me, the Toolik snowpack is more similar to a polar than a temperate snowpack (e.g., given permanent darkness/sunlight periods).

3. Additionally, there is no discussion on instrumental/analytical uncertainties. I would for instance very much like to know how the results collected with the two Tekran instruments compare. I expect a 10-20 % difference and wonder if the conclusions remain valid taking that analytical uncertainty into account. I also wonder whether the upper inlet in the snowpack was too far from the surface in spring, explaining why the authors did not observe any photochemical production of Hg(0)gas in the upper layers.

The following line by line comments should be useful to fully comprehend and address the major comments.

Line by line comments

Page 2, lines 21-22: "In the Arctic and Antarctic, Hg cycling also is affected by atmospheric Hg depletion events (AMDEs) which are observed primarily in the springtime along coastal locations (Dommergue et al., 2010; Schroeder et al., 1998; Steffen et al., Interactive comment

2008)." The authors should also cite Angot et al. (2016a), latest review paper about mercury in polar regions which includes a discussion on the occurrence and frequency of AMDEs in recent years.

Page 2, line 28: Typo; McMurod should be McMurdo.

Page 4, lines 6-8: "Atmospheric air sampling was performed using the top snow tower air inlet (...), as well as on a nearby micrometeorological tower at a height of 3.6 m above ground." How do the results compare? Did you collect data with two different Tekran instruments? If so, it would be an easy way to answer the following comment.

Page 4, lines 10-11: "Gaseous Hg(0) concentrations were measured using two Tekran 2537B analyzers, one for interstitial snow air measurements and the other shared for soil gas and atmospheric measurements." Did you check how the results from the two instruments compare? What is the associated uncertainty? According to several studies, the analytical uncertainty is about 10-20 % (e.g., Slemr et al., 2015) and this should be taken into account when comparing data acquired with different instruments. See major comments.

Page 4, line 12: "Air sampling was alternated between different snowpack heights every 5 min so that a full sequence of air extraction from the snowpack (six inlet heights) was achieved every 30 minutes." If I understood correctly, you just collected one data point per inlet height. Could there be any significant sampling-induced snowpack ventilation influencing the results?

Page 4, line 27: "The top 3 cm of the snowpack was collected in triplicate."

1. What was the approximate distance between the replicates? According to lines 3-5 on page 6, the Toolik snowpack is subject to significant drifts and changes in snowpack height. Did you take that into account when interpreting the transect data? See major comments.

2. According to lines 30-31 on page 5, average data are shown as mean ïĆś standard

TCD
deviation. Were the replicate surface snow samples pooled together for analysis, similarly to snowpit samples? If so, what does the standard deviation reported in Table S2 stand for?

Page 5, lines 7-9: "The detection limits, determined as 3-times the standard deviation of blank samples, averaged 0.08 ng/L. For statistic purpose, values below the detection limit (DL) were included as 0.5xDL. Recoveries, determined by 5 ng/L standards analyzed every 10 samples, averaged between 93 and 107 %." If I understood correctly, the analytical uncertainty is of 7 % at 5 ng/L. Given that most of the concentrations you analyzed are

concentrations in soils (mostly below the detection limits for both years, i.e.,

an underestimation of Hg(II) concentration conceivable?

b) Since Toolik snowpack is subject to significant drifts and changes in snowpack height, are you 100 % sure that you collected surface snow samples? If you collected deeper layers due to drifting snow that could explain the unusually low concentrations during the depletion event.

Page 7, lines 32-34: "This pattern was consistent for two independent soil profiles measured at this site, one mainly representing an organic soil profile and one profile dominated by mineral soil horizons". I do not understand this sentence. Was the experimentation carried out at two different locations with different soil composition? There is no mention of this in the Materials and Methods section.

Page 8, lines 25-27: "Analysis of ïĄĎHg/ïĄĎCO2 ratios showed no statistically significant differences from the top to the bottom of the snowpack". It seems like the authors used the mean concentration at each height to calculate the ïĄĎHg/ïĄĎCO2 ratios. Given the quite large error bars for Hg(0)gas in the snowpack (Fig. 4), it is not really surprising that such a calculation yields insignificant differences. On the contrary, Fain et al. (2013) calculated the ratio for each day. Did you try to do it this way?

Page 9, lines 7-10: "No consistent temporal trends in Hgtot or Hgdiss were observed with increasing duration of winter in both seasons, and no correlations were observed with air temperature (red line). One noticeable period of enhanced surface snow Hg concentrations was April 2016 when both Hgtot and Hgdiss concentrations exceeded 1 ng/L."

1. Is there an anti-correlation between Hgtot and Hg(0) in the atmosphere?

2. See previous comments regarding the magnitude of the concentrations. How do a maximum of 1 ng/L compare with other studies performed in polar regions?

Page 9, line 12: "Due to the low snow height on the frozen lake." Why is the snow height lower on the frozen lake? Drifting snow?
Page 9, lines 15-16: "Measurements of Hgtot and Hgdiss across a large North slope transect (about 200 km) in March, 2016 showed concentrations of 0.70 ïĆś 0.79 and 0.24 ïĆś 0.20 ng/L, respectively." Are you referring to surface snow or to average snowpack concentrations?

Page 9, lines 16-19: "Concentrations of Hgdiss of the 5 northernmost stations were statistically significantly higher compared to those measured in the 4 stations located in the interior tundra which included the Toolik site where the mean Hgdiss concentrations were 0.33 and 0.11 ng/L for the same period, respectively".

1. Please add the standard deviations (in the text and on Figure 7)

2. I would like to see a critical discussion of these results, notably in light of drifting snow issues and analytical uncertainty (50 % at such low concentrations?).

Page 9, line 23: "similar to concentration of Hg". Where is that discussed in the manuscript?

Page 11, lines 11-12: "Hg(0) concentration profiles in the arctic snowpack are inherently different to patterns observed in lower latitude snowpacks". Now may be a good time to compare your results to other studies in polar regions (see comment page 7, line 3).

Page 11, lines 23-25: "We speculate that a reason for the general lack of Hg(0)gas formation and volatilization in snow includes substrate limitation due to very low total snow Hg concentrations, several times lower compared to concentrations in temperate snowpacks." While I agree that this is a possible explanation, I would like to see a more thorough discussion of other conceivable hypothesis, including instrumental limitations. A few ideas:

1. The comparison between your results and those by Fain et al. (2013) is based on data acquired in winter and early spring. I would expect Hg(0)gas formation and volatilization to start in spring, when the sun is back. What is the difference in UV Interactive comment

load between Toolik and Colorado at that time of year? That could explain a lower photo-reduction in the upper layers of the snowpack at Toolik.

2. Atmospheric and snowpack Hg(0) concentrations were collected using two different Tekran instruments. Could a 10-20 % difference between the instruments explain why Hg(0)gas in surface snow is not higher than in ambient air (especially Figure 4c)?

3.  $\sim$ 85 % of photo-reduction occurs in the top two e-folding depths (King and Simpson, 2001). Fain et al. (2013) observed a diurnal cycle 0-60 cm below the surface. But the density/nature of the snowpack is most likely very different at Toolik.

a) You recorded the physical properties (e.g., density) of the snowpack. Could you compare your results to those obtained by Fain et al. (2013)? According to Durnford and Dastoor (2011), the range of chemically active depths may be explained by physical differences in the snowpacks.

b) The e-folding depth is about 5 cm at Alert (King and Simpson, 2001) while Poulain et al. (2004) reported that photo-reduction occurred in the first 3 cm of snow. I am worried that the upper inlet in the snowpack (the one at  $\sim$ 40 cm above the ground, Fig. 4c) might be too far from the surface. This is more obvious on Fig.3 if you compare the distance between the upper inlet and the surface in your case and in Fain et al. (2013).

4. What about the occurrence of fresh snow at Toolik vs. Colorado? Fresh snow provides a new reservoir of photoreducible Hg(II) and highest surface snow Hg(0) levels are linked to the deposition of new snow (Faïn et al., 2013).

Page 12, lines 1-2: "This illustrates that the Hg(0)gas uptake occurs in soils rather than in the snowpack." The way the ratios are plotted, I am not sure I understand why you can conclude that uptake occurs in soils rather than in the snowpack.

Page 12, line 10-11: "which we attribute to higher variability in upper snowpack concentrations due to variable atmospheric Hg(0)gas levels". Isn't it in contradiction with error bars (Fig. 4 and 5)? I actually don't understand why (Fig 5) a ratio based on TCD
highly variable concentrations (e.g., 0 cm) can be less variable than a ratio based on less variable concentrations (e.g., upper inlet). Am I missing something?

Page 12, line 23: "relatively weak and infrequent AMDEs". I quibble but the AMDE reported on Fig. S2 does not look weak to me. About the frequency, how many AMDEs did you observe? You can use a 1.00 ng/m3 threshold to calculate the frequency of occurrence (Angot et al., 2016a; Cobbett et al., 2007; Steffen et al., 2005).

Page 12, line 31-32: "Total Hg concentrations in all snow samples collected were always much higher than Hgdiss levels". Can we really say that concentrations are "much higher" given the range observed (0-1 ng/L)? By the way, I don't see anywhere Hgtot concentrations from the transect.

Page 13, line 10: "(...) and few studies include inland sites such as Toolik". What is the range of concentrations at these inland sites? Which studies are you referring to?

Page 14, line 27: "Fresh surface snow". There is no mention of the fact that you collected fresh surface snow in the Materials and Methods Section.

Figure 2: Why don't you report the standard deviation for each green bar?

Figure 4: What does "soil: organic/mineral" refer to?

Figure 7: Please add standard deviations.

Figure 8: I really like this Figure but aren't the various concentrations (

Pirrone, N., Ryjkov, A., Selin, N.E., Skov, H., Song, S., Sprovieri, F., Steffen, A., Toyota, K., Travnikov, O., Yang, X., Dommergue, A., 2016a. Chemical cycling and deposition of atmospheric mercury in polar regions: review of recent measurements and comparison with models. Atmos Chem Phys 16, 10735–10763. https://doi.org/10.5194/acp-16-10735-2016

Angot, H., Magand, O., Helmig, D., Ricaud, P., Quennehen, B., Gallée, H., Del Guasta, M., Sprovieri, F., Pirrone, N., Savarino, J., Dommergue, A., 2016b. New insights into the atmospheric mercury cycling in central Antarctica and implications on a continental scale. Atmos Chem Phys 16, 8249–8264. https://doi.org/10.5194/acp-16-8249-2016

Cobbett, F.D., Steffen, A., Lawson, G., Van Heyst, B.J., 2007. GEM fluxes and atmospheric mercury concentrations (GEM, RGM and Hg(p)) in the Canadian Arctic at Alert, Nunavut, Canada (February–June 2005). Atmos. Environ. 41, 6527–6543. https://doi.org/10.1016/j.atmosenv.2007.04.033

Durnford, D., Dastoor, A., 2011. The behavior of mercury in the cryosphere: A review of what we know from observations. J. Geophys. Res. Atmospheres 116, D06305. https://doi.org/10.1029/2010JD014809

Faïn, X., Helmig, D., Hueber, J., Obrist, D., Williams, M.W., 2013. Mercury dynamics in the Rocky Mountain, Colorado, snowpack. Biogeosciences 10, 3793–3807. https://doi.org/10.5194/bg-10-3793-2013

King, M.D., Simpson, W.R., 2001. Extinction of UV radiation in Arctic snow at Alert, Canada (82°N). J. Geophys. Res. Atmospheres 106, 12499–12507. https://doi.org/10.1029/2001JD900006

Obrist, D., Agnan, Y., Jiskra, M., Olson, C.L., Colegrove, D.P., Hueber, J., Moore, C.W., Sonke, J.E., Helmig, D., 2017. Tundra uptake of atmospheric elemental mercury drives Arctic mercury pollution. Nature 547, 201–204. https://doi.org/10.1038/nature22997

Poulain, A.J., Lalonde, J.D., Amyot, M., Shead, J.A., Raofie, F., Ariya, P.A., 2004. Re-
dox transformations of mercury in an Arctic snowpack at springtime. Atmos. Environ. 38, 6763–6774. https://doi.org/10.1016/j.atmosenv.2004.09.013 Slemr, F., Angot, H., Dommergue, A., Magand, O., Barret, M., Weigelt, A., Ebinghaus, R.,

Brunke, E.-G., Pfaffhuber, K.A., Edwards, G., Howard, D., Powell, J., Keywood, M., Wang, F., 2015. Comparison of mercury concentrations measured at several sites in the Southern Hemisphere. Atmos Chem Phys 15, 3125–3133. https://doi.org/10.5194/acp-15-3125-2015

Steffen, A., Bottenheim, J., Cole, A., Ebinghaus, R., Lawson, G., Leaitch, W.R., 2014. Atmospheric mercury speciation and mercury in snow over time at Alert, Canada. Atmos Chem Phys 14, 2219–2231. https://doi.org/10.5194/acp-14-2219-2014

Steffen, A., Schroeder, W., Macdonald, R., Poissant, L., Konoplev, A., 2005. Mercury in the Arctic atmosphere: An analysis of eight years of measurements of GEM at Alert (Canada) and a comparison with observations at Amderma (Russia) and Kuujjuarapik (Canada). Sci. Total Environ., Sources, Occurrence, Trends and Pathways of Contaminants in the ArcticBidleman S.I. 342, 185–198. https://doi.org/10.1016/j.scitotenv.2004.12.048

TCD

---

## Referee Comment (RC2) · Anonymous Referee #2 · 7 Mar 2018

**1 General Comments**

The manuscript describes a detailed study of Hg in air, snow and soil at an Arctic site, which unlike almost all previous studies is a significant distance from the coast. Given that vast amounts of tundra are inland this study begins to fill in some of the gaps in our knowledge of Hg cycling in these remote regions. Of particular interest are the differences seen between the processes seen at this site when compared to coastal sites. The fact that tundra soils are a sink for atmospheric elemental Hg has important repercussions for future multimedia modelling studies and hints at the potential remobilisation of large amounts of Hg from Arctic soils in a warming climate. This should be

emphasised more in the Abstract and the Conclusions, in the Abstract particularly the comment on this is brief and hidden in the middle.

The manuscript is however rather long-winded. I think that both the Results and the Discussion section could be shortened significantly, and quite usefully (from the reader's point of view) combined. Just as an example, the discussion of the major ions and the O and H isotope signatures, repeats parts of the pertinent results section. Conversely the results section rather leaves the reader with a sense of 'and what do these results imply?', which is only answered six pages later. I would recommend combing these sections as it will most likely lead naturally to a more concise and less prolix article.

If some of the detail in the methods section has already been published perhaps it could be shortened by including more references, if not maybe some of the detail could be moved to the Supplementary material.

The previous reviewer has comprehensively addressed a number of technical issues, and for me only a few real problems remain.

1. The issue of blowing snow, and where the snow that is being sampled at Toolik comes from.

The fact that the paper is interesting and adds an important contribution to polar Hg research but is unfortunately not very well presented and at times rather heavy going.
The importance of atmospheric elemental Hg effectively being sequestered (for the moment) by tundra soils, is not emphasised sufficiently from my point of view.

TCD

---

## Author Comment (AC1) · 19 Apr 2018

**REVIEWER 1**

**General comments**

The authors investigated snow Hg dynamics in the interior arctic tundra at Toolik, Alaska. They compared their results to a temperate snowpack in the Rocky Mountain, Colorado and conclude that photochemical Hg(0) losses from the snowpack in models should be adjusted and treated differently in arctic and temperate snowpacks. I think that this research provides some very interesting results and the manuscript is fairly well written. However, to my opinion, it lacks a thorough comparison to other studies performed in polar regions and a discussion on instrumental limitations. I think that the manuscript will be appropriate for publication in The Cryosphere after the authors address the major comments discussed below.

**Major comments**

The conclusion of the manuscript (see above) derives from two main observations. Firstly, low concentrations of Hgtot and Hgdiss were measured in surface snow. Secondly, the photochemical formation of Hg(0)gas in the snowpack was largely absent.

1. I am quite surprised by the very low Hgtot concentrations in surface snow samples reported in this study, especially during depletion events. This raises concern about the potential influence of drifting snow. While the authors admit that the Toolik snowpack is subject to significant drifts and changes in snowpack height, there is no discussion on potential consequences and uncertainties.

We added more discussions about that. See below for more details. However, since we did not observe Hg concentration trend in snowpacks with depths or differences between lake and tundra snowpacks, we largely focus our discussion on the importance of spatial redistribution on snowpack Hg pool sizes.

2. While I agree that the photochemical formation of Hg(0)gas in the snowpack is most likely low compared to a temperate snowpack, I would like to see a comparison to other studies performed in polar regions. To me, the Toolik snowpack is more similar to a polar than a temperate snowpack (e.g., given permanent darkness/sunlight periods).

We added additional comparisons and discussions with respect to other polar snowpack (see below). However, we maintain our discussion and comparison to temperate snowpack as well to highlight the dramatic difference between temperate and arctic snowpacks.

3. Additionally, there is no discussion on instrumental/analytical uncertainties. I would for instance very much like to know how the results collected with the two Tekran instruments compare. I expect a 10-20 % difference and wonder if the conclusions remain valid taking that analytical uncertainty into account. I also wonder whether the upper inlet in the snowpack was too far from the surface in spring, explaining why the authors did not observe any photochemical production of Hg(0)gas in the upper layers.

Please note that the Tekran used for the snowpack tower had one inlet line in the atmosphere which allowed us to adjust the two Tekrans used in this study based on atmospheric measurements (see detailed comment below). So, the differences between atmospheric and snowpack Hg concentrations cannot be explained by instrument variability.

The uppermost inlet in snow was always ≤10 cm from the atmosphere (note that was the spacing between snow inlets). Also, the absolute inlet depth (from the top of the snowpack) is variable based on the snowpack height. We added a discussion point that a possible reason for not detecting Hg$^0_{gas}$ production may be that if this process was limited to a few cm in the top snow, our measurements would have missed this. We also determined using a webcam and snowpack sticks that the upper snowpack inlet was on average 7 cm from the atmosphere during the Apr. 2016's AMDE, 5 cm in Mar. 2016, and 6 cm in Mar. 2015 (i.e., during for the three biggest AMDEs). Also, we added that when using the exactly same measurement system, detecting Hg$^0_{gas}$ enhancements in temperate snowpack (Fain et al., 2013) apparently were not problematic using this system.

The following line by line comments should be useful to fully comprehend and address the major comments.

**Line by line comments**

Page 2, lines 21-22: "In the Arctic and Antarctic, Hg cycling also is affected by atmospheric Hg depletion events (AMDEs) which are observed primarily in the springtime along coastal locations (Dommergue et al., 2010; Schroeder et al., 1998; Steffen et al., 2008)." The authors should also cite Angot et al. (2016a), latest review paper about

mercury in polar regions which includes a discussion on the occurrence and frequency of AMDEs in recent years.

Thanks for this reference. We added it.

Page 2, line 28: Typo; McMurod should be McMurdo.

Done, thanks!

Page 4, lines 6-8: "Atmospheric air sampling was performed using the top snow tower air inlet (. . .), as well as on a nearby micrometeorological tower at a height of 3.6 m above ground." How do the results compare? Did you collect data with two different Tekran instruments? If so, it would be an easy way to answer the following comment.

We collected data from two different Tekran instruments. The first one was connected to the micrometeorological tower (data not used in this paper, but in Obrist at al., 2017) and the soil wells (used in this paper). The second one was connected to the snow tower (snowpack measurement by 0, 10, 20, 30, and 40-cm inlets, as well as atmospheric levels measured at the top inlet of the snowpack (110-cm above ground, hence always located in the atmosphere). For this study, we hence always compared the snow data to atmospheric data using the same Tekran analyzer. For further comparisons, we were also able to adjust atmospheric measurements of the Tekran connected to the snow tower with atmospheric gradients: (we found on average 7% of differences, which could be adjusted to eliminated this difference). We added some information in the text to clarify this on page 4 lines 14-19.

Page 4, lines 10-11: "Gaseous Hg(0) concentrations were measured using two Tekran 2537B analyzers, one for interstitial snow air measurements and the other shared for soil gas and atmospheric measurements." Did you check how the results from the two instruments compare? What is the associated uncertainty? According to several
studies, the analytical uncertainty is about 10-20 % (e.g., Slemr et al., 2015) and this should be taken into account when comparing data acquired with different instruments. See major comments.

See responses above.

Page 4, line 12: "Air sampling was alternated between different snowpack heights every 5 min so that a full sequence of air extraction from the snowpack (six inlet heights) was achieved every 30 minutes." If I understood correctly, you just collected one data point per inlet height. Could there be any significant sampling-induced snowpack ventilation influencing the results?

We apologize that there was a mistake in this description. We clarified that snow tower measurements were based on switching heights every 10-min corresponding to two individuals of 5-min measurements per inlet (to avoid Tekran trap bias), resulting in 1-hour measurements per full gradient. The reviewer raises an important point, that is if and to what degree the active sampling of air from the snowpack induces artificial ventilation and dilution of the natural gas profile. The dual inlet sampling at a given depth, the 90° rotation of inlets at adjacent depths, and the only intermittent sampling (10 min every 1 h) were all selected to minimize this effect. The higher resolution trace gas ($CO_2$, $O_3$) were used to assess this artifact.
Concentration changes within the 10-min interval were relatively minor (<10–20%) which suggests that mostly air from within the vicinity of the inlets was sampled. We have thoroughly evaluated this in other previous descriptions of this snowpack air sampling approach and refer the reviewer to those publications for a more in depth assessment (Seok et al., 2009). Moreover, our continuous and automatic sampling (i.e., resulting in 24 full snowpack profiles each day when data coverage is complete), limits any systematic bias that would be introduced by ventilation.

Page 4, line 27: "The top 3 cm of the snowpack was collected in triplicate."
1. What was the approximate distance between the replicates? According to lines 3-5 on page 6, the Toolik snowpack is subject to significant drifts and changes in snowpack height. Did you take that into account when interpreting the transect data? See major comments.

The triplicate samples were collected within a distance of 5 m (we added this in the text). We do not focus our discussion on surface snow patterns (since there was no significant difference to snowpack-averaged data with the exception of active AMDEs periods), so we did not further evaluate/discuss how drifts in surface snow may affect results.

2. According to lines 30-31 on page 5, average data are shown as mean ï˙C ˙s standard deviation. Were the replicate surface snow samples pooled together for analysis, similarly to snowpit samples? If so, what does the standard deviation reported in Table S2 stand for?

    Surface snow samples were analyzed separately. Surface snow SD is based on the triplicates (n=3) while snowpack SD is based on the two replicate pits (n=2). The two snowpack samples pooled together were collected from the same pit, but from different walls. We added this information in the Table S2 caption and clarified this in the methods (from page 4 line 25).

Page 5, lines 7-9: "The detection limits, determined as 3-times the standard deviation of blank samples, averaged 0.08 ng/L. For statistic purpose, values below the detection limit (DL) were included as 0.5xDL. Recoveries, determined by 5 ng/L standards analyzed every 10 samples, averaged between 93 and 107 %." If I understood correctly, the analytical uncertainty is of 7 % at 5 ng/L. Given that most of the concentrations you analyzed are < 1.0 ng/L, did you check what the analytical uncertainty at that concentration is? I expect it to be much higher (_50%?). Additionally, is the analytical uncertainty the same for Hgtot and Hgdiss? I am asking because you say later (page 13, lines 1-2) that Hgtot concentrations were much more variable than Hgdiss concentrations.

    This is an excellent point. We indeed determined analyzer performance using recoveries of 5 ng $L^{-1}$ samples every 10 samples, and report a recovery between 93 and 107% for these samples. We did not assess the stability of the analyzer specifically at concentrations <1.0 ng $L^{-1}$, and this could be more variable. However, it is unlikely that higher variability in $Hg_{tot}$ can be explained by this since concentrations are higher in $Hg_{tot}$ than $Hg_{diss}$.

Page 6, lines 24-25: "The transect between Toolik and the Arctic Ocean performed in March 2016 showed snowpack height ranging between 30 and 66 cm". How can you explain the difference? Could it be due to drifting snow? If so, how can you compare Hg concentrations at various depths and locations? See major comment.

    We believe (and clarified) that such differences reflect a high degree of variability in snowpack distribution across the area, but our dataset is not dense enough (or frequent enough) to explain differences in depth across locations in this study. We state, however, that we observed that snow height is higher at Toolik (between 40 and 50 cm) compared to coastal sites (<30 cm). Secondly, we state that highest snow height was observed where the vegetation was composed of shrubs which may have facilitated snow accumulation.
    We also focused our spatial discussion on pool size of Hg stored in the snowpack because a detailed comparison of depth patterns would require a frequent and detailed observations of snowpack drifts which is not possible in our study. We clarified this point.

Page 7, line 3: "compared to the literature from temperate snowpacks". It seems to me that your reference is the study by Fain et al. (2013) in Colorado. "compared to the literature from a temperate snowpack" would be more appropriate here. Additionally, while I understand why you compare your results to those obtained in the Rocky Mountain (same instrumental setup and so on), I would like to see a more thorough comparison to other studies performed in polar regions (e.g., Angot et al., 2016b; Steffen et al., 2014).

    We perform a detailed comparison to Faïn et al. (2013) because that study used the same instrumental setup. However, we added a sentence for comparing $Hg^0_{gas}$ data to other polar regions (see page 8, lines 3-7).

Page 7, lines 10-12: "The Hg(0)gas measurements consistently showed strong concentration gradients in the atmosphere-snowpack-soil continuum with highest concentrations in the atmosphere (on average, 1.18 and 1.09 ng/m3, respectively) and lowest concentrations in soils (mostly below the detection limits for both years, i.e., <0.05 ng/m3)."
1. Could you please add the standard deviations?

    Done!

2. Is this gradient significant given the large error bars in the snowpack (see Fig. 4)?

    We clarified that the high error bars shown in Fig. 4 are mainly due to the temporal variability. The gradient is well defined for each given measurement sequence (see Fig. 1 in Obrist et al., 2017).

3. It's just a detail, but I think that the detection limit for the Tekran 2537 is 0.10 ng/m3.

Thanks for this remark, indeed, it is <0.10 ng m$^{-3}$ for the Tekran 2537. We corrected that.

Page 7, lines 23-25, referring to AMDEs: "During one of these periods shown in Figure S2, Hg(0)gas concentrations in the snowpack showed variable Hg(0)gas levels generally following Hg(0)gas concentration changes in the atmosphere above".
1. What do you mean by "generally following"? Is there a correlation between concentrations in the atmosphere and upper layers of the snowpack?

Hg$^0_{gas}$ concentration variations in the upper layers (as well as for deeper layers) are linked to the Hg$^0_{gas}$ concentration variations in the atmosphere (due to the snowpack diffusivity). We clarify this point in the manuscript (page 8, lines 27-31).

2. Could you please add the following data on Figure S2?
a) Ozone in the atmosphere b) Snow height (is the inlet at 20 cm above the ground far from the surface?) c) Hgtot in surface snow samples d) Hg(II) concentrations in the atmosphere (the data do exist according to Obrist et al. (2017)).

O$_3$ and Hg$^{II}$ concentrations were added in the figure. The snow height was between 25 and 28 cm, i.e., 5 to 8 cm far from the surface (information added in the caption). Hg$_{tot}$ concentrations in surface snow samples were 1.00±0.07 ng L$^{-1}$ for Mar. 25$^{th}$ and 1.46±0.16 ng L$^{-1}$ for Apr. 2$^{nd}$ (only concentrations that we have, available in Table S2).

3. How can you explain the peaks at 10 cm above the ground (e.g., daytime on Mar 28th, 29th and 30th)? Was the temperature in the snowpack or the sample line stable? Could it be Hg(II) released from the sample line and analyzed as Hg(0)?

In general, we found similar diurnal patterns in the atmosphere as well as in all snowpack levels. We attribute such diurnal variability to differences in daytime and nighttime boundary layer mixing. However, we cannot specifically explain why inlet depth at 10 cm was more variable during this time. In general, atmospheric Hg$^0_{gas}$ concentrations, which then also affect snowpack concentrations, are most variable during these periods of AMDEs.

4. According to table S2 you collected surface snow samples during the AMDE. I am very surprised by the very low concentration (1.46 ng/L). This is rather unusual during a depletion event. How does it compare to other studies (e.g., Steffen et al., 2014)? If we do a back of the envelope calculation based on Hg(0) and Hg(II) concentrations in the atmosphere, what should be the concentration in surface snow to have a coherent Hg budget? With 1.46 ng/L in surface snow and _0.4 ng/m3 of Hg(II) according to Obrist et al. (2017), I have the feeling that there is Hg missing in the budget. If so, a) How reliable are the Hg(II) measurements? What is the analytical uncertainty? Is an underestimation of Hg(II) concentration conceivable? b) Since Toolik snowpack is subject to significant drifts and changes in snowpack height, are you 100 % sure that you collected surface snow samples? If you collected deeper layers due to drifting snow that could explain the unusually low concentrations during the depletion event.

Indeed, the concentrations are low compared to Steffen et al. (2014) data. We revised this discussion and now state that the coarse resolution of our snow sampling campaign during AMDEs is not sufficient to closely track the fate of Hg deposition and subsequent Hg reemissions during AMDEs. So, while we feel that the reviewer's points are excellent, we strongly think a mass balance approach is above what we are comfortable to address in this study with a very coarse snow sampling activities during AMDEs.
However, if we were to perform a back-of the envelope mass balance, it would amount to the following: First, please note that during the weeks of the most intense AMDEs (last week of March and first week of April), we observed highly variable daily average atmospheric Hg$^{II}$ concentrations, ranging between non-detectable levels to a daily average of 310 pg m$^{-3}$ based on the dataset presented in Obrist et al. (2017) (Extended Data Figure 2). Hence, peak Hg$^{II}$ concentrations of ~400 pg m$^{-3}$ as the reviewer refers were only observed during very few hourly measurements and cannot be applied over long time. In fact, we estimated a total Hg$^{II}$ deposition of between 0.8 and 2.8 µg m$^{-2}$ a$^{-1}$ (based on a deposition velocity of 1.5 cm s$^{-1}$), of which most of this occurred during periods of AMDEs. At the same time, measured Hg$^0_{gas}$ fluxes (re-emissions) showed a similar magnitude of Hg$^0_{gas}$ re-emissions, e.g., of 1.5 µg m$^{-2}$ in ~20 day during AMDEs (Figure 1 of Obrist et al., 2017). Hence, we believe in fact that re-emission of Hg$^0_{gas}$ after AMDEs largely accounted for the estimated Hg$^{II}$ deposition, a finding we support with stable Hg isotope measurements in Obrist et al. (2017).

Now, looking at surface snow (3 cm, which has a water equivalent of 10 L m$^{-2}$ considering a snow density of 0.3), an increase in snow Hg (difference of Hg$_{diss}$ from 0.26 to 1.15 = 0.89 ng L$^{-1}$) would represent a very small input of only ~9 ng m$^{-2}$ during the AMDEs. So, in fact, what would find that snow indeed only represent a very low amount of the potential Hg$^{II}$ deposition and Hg$^0_{gas}$ re-emission. It would be interesting to perform a high-resolution mass balance campaign to address this issue further during AMDEs.

Page 7, lines 32-34: "This pattern was consistent for two independent soil profiles measured at this site, one mainly representing an organic soil profile and one profile dominated by mineral soil horizons". I do not understand this sentence. Was the experimentation carried out at two different locations with different soil composition? There is no mention of this in the Materials and Methods section.

We clarified that of our soil measurements, we had measurements in different soils types (present within 5 m), some were more organic soils a some were more mineral soils.

Page 8, lines 25-27: "Analysis of ïA˛D˘ Hg/ïA˛D˘ CO2 ratios showed no statistically significant differences from the top to the bottom of the snowpack". It seems like the authors used the mean concentration at each height to calculate the ïA˛D˘ Hg/ïA˛D˘ CO2 ratios. Given the quite large error bars for Hg(0)gas in the snowpack (Fig. 4), it is not really surprising that such a calculation yields insignificant differences. On the contrary, Fain et al. (2013) calculated the ratio for each day. Did you try to do it this way?

Actually, the ratios presented in boxplots of the Fig. 4 were in fact calculated from daily Hg$^0_{gas}$ and CO$_2$ measurements, as performed by Faïn et al. (2013). See below the detail on the individual days. On the left panel, each dataline represents each day (from darker to lighter). On the right panel, the same plot without 3 outliers. No apparent trend was observed. What we have done in Fig. 4 was to apply boxplot for each snow height difference (0 to 10, 10 to 20, and 20 to 30 cm).

[Figure]

Page 9, lines 7-10: "No consistent temporal trends in Hgtot or Hgdiss were observed with increasing duration of winter in both seasons, and no correlations were observed with air temperature (red line). One noticeable period of enhanced surface snow Hg concentrations was April 2016 when both Hgtot and Hgdiss concentrations exceeded 1 ng/L."

1. Is there an anti-correlation between Hgtot and Hg(0) in the atmosphere?

No, there is no anti-correlation between Hg$_{tot}$ and Hg$^0_{gas}$, nor for Hg$_{diss}$ and Hg$^0_{gas}$, during most of the time. One exception was during AMDE where we observed low Hg$^0_{gas}$ and higher snow Hg concentrations.

2. See previous comments regarding the magnitude of the concentrations. How do a maximum of 1 ng/L compare with other studies performed in polar regions?

We indicated that snowpack Hg$_{tot}$/Hg$_{diss}$ measured in literature were between 0.14 and 820 ng L$^{-1}$. This range mostly includes Hg$_{tot}$ concentrations (and so, potentially higher concentrations) when no filtration was applied. We state that our observed concentrations were in the lower part of the Arctic data set: the lowest values were observed in Greenland (Ferrari et al., 2004), in Canada for Hg$_{tot}$ (St Louis et al., 2005), and in Alaska for Hg$_{diss}$ (Douglas & Sturm, 2004).

Page 9, line 12: "Due to the low snow height on the frozen lake." Why is the snow height lower on the frozen lake? Drifting snow?

We assume that both drifting snow and surface roughness (brush vs ice) are responsible to the difference of snow height, as already discussed page 11, from line 4.

Page 9, lines 15-16: "Measurements of Hgtot and Hgdiss across a large North slope transect (about 200 km) in March, 2016 showed concentrations of 0.70 ï´C´s 0.79 and 0.24 ï´C´s 0.20 ng/L, respectively." Are you referring to surface snow or to average snowpack concentrations?

We referred here to average snowpack concentrations. Surface snow sampling was not performed for the transect. We clarified that.

Page 9, lines 16-19: "Concentrations of Hgdiss of the 5 northernmost stations were statistically significantly higher compared to those measured in the 4 stations located in the interior tundra which included the Toolik site where the mean Hgdiss concentrations were 0.33 and 0.11 ng/L for the same period, respectively".
1. Please add the standard deviations (in the text and on Figure 7)

The standard deviations were added. The standard deviations of the Fig. 7, however, were not added since only a composite sample from both replicates of the same snow height was measured.

2. I would like to see a critical discussion of these results, notably in light of drifting snow issues and analytical uncertainty (50 % at such low concentrations?).

We discussed the influence of drifting snow in each part of the discussion (absence of vertical patter, absence of difference between the lake and tundra snowpacks…). Note that we reorganized the manuscript merging Results and Discussion.

Page 9, line 23: "similar to concentration of Hg". Where is that discussed in the manuscript?

We clarified this sentence. Major cations and anions concentrations in snowpack were lower at Toolik compared to coastal locations, which we also observed for Hg as discussed in the previous section.

Page 11, lines 11-12: "Hg(0) concentration profiles in the arctic snowpack are inherently different to patterns observed in lower latitude snowpacks". Now may be a good time to compare your results to other studies in polar regions (see comment page 7, line 3).

We added the comparison with other polar snowpacks, as discussed above.

Page 11, lines 23-25: "We speculate that a reason for the general lack of Hg(0)gas formation and volatilization in snow includes substrate limitation due to very low total snow Hg concentrations, several times lower compared to concentrations in temperate snowpacks." While I agree that this is a possible explanation, I would like to see a more thorough discussion of other conceivable hypothesis, including instrumental limitations. A few ideas:
1. The comparison between your results and those by Fain et al. (2013) is based on data acquired in winter and early spring. I would expect Hg(0)gas formation and volatilization to start in spring, when the sun is back. What is the difference in UV load between Toolik and Colorado at that time of year? That could explain a lower photo-reduction in the upper layers of the snowpack at Toolik.

Indeed, the solar radiation is a little bit lower at Toolik. We clarified that solar radiation at Toolik was about 400 W m$^{-2}$ in Mar. and 600 W m$^{-2}$ in Apr. at Toolik versus ~700 W m$^{-2}$ in the end of Feb. in the Rocky Mountain (Faïn et al., 2013). The solar radiation data were added in the text (page 8, lines 17-19). However, we do not believe this can explain the dramatically different behavior between these sites.

2. Atmospheric and snowpack Hg(0) concentrations were collected using two different Tekran instruments. Could a 10-20 % difference between the instruments explain why Hg(0)gas in surface snow is not higher than in ambient air (especially Figure 4c)?

See our comment above. In this study, we always compared snow and atmospheric measurements using the same Tekran instrument.

3. 85 % of photo-reduction occurs in the top two e-folding depths (King and Simpson, 2001). Fain et al. (2013) observed a diurnal cycle 0-60 cm below the surface. But the density/nature of the snowpack is most likely very different at Toolik.

a) You recorded the physical properties (e.g., density) of the snowpack. Could you compare your results to those obtained by Fain et al. (2013)? According to Durnford and Dastoor (2011), the range of chemically active depths may be explained by physical differences in the snowpacks.

b) The e-folding depth is about 5 cm at Alert (King and Simpson, 2001) while Poulain et al. (2004) reported that photo-reduction occurred in the first 3 cm of snow. I am worried that the upper inlet in the snowpack (the one at 40 cm above the ground, Fig. 4c) might be too far from the surface. This is more obvious on Fig.3 if you compare the distance between the upper inlet and the surface in your case and in Fain et al. (2013).

We added the measurement depths of the surface snow inlet which was between 5 and 7 cm. We also discuss the possibility that our surface measurements may have missed $Hg^0_{gas}$ production. However, at the same point, we point out that in Fain et al. (2013), we used the same measurement system and that $Hg^0_{gas}$ concentration enhancements in that study were well measurable (with concentrations up to 8 ng m$^{-3}$ and detectable to a depth of >90 cm).

4. What about the occurrence of fresh snow at Toolik vs. Colorado? Fresh snow provides a new reservoir of photoreducible Hg(II) and highest surface snow Hg(0) levels are linked to the deposition of new snow (Faïn et al., 2013).

We performed plot as done in Fig. 5 in Faïn et al. (2013) but no relationships were observed.

Page 12, lines 1-2: "This illustrates that the Hg(0)gas uptake occurs in soils rather than in the snowpack." The way the ratios are plotted, I am not sure I understand why you can conclude that uptake occurs in soils rather than in the snowpack.

We clarified the results as follows: the constant ratios and the fact that $CO_2$ is largely non-reactive in snowpack indicates that $Hg^0_{gas}$ also was not subject to snowpack chemical reactions. The constant, and negative ratios, between $CO_2$ and $Hg^0_{gas}$ ratios are hence indicative that both profiles are affected by underlying soil processes, i.e., soil sources for $CO_2$ and soil sinks (for $H^0_{gas}$).

Page 12, line 10-11: "which we attribute to higher variability in upper snowpack concentrations due to variable atmospheric Hg(0)gas levels". Isn't it in contradiction with error bars (Fig. 4 and 5)? I actually don't understand why (Fig 5) a ratio based on highly variable concentrations (e.g., 0 cm) can be less variable than a ratio based on less variable concentrations (e.g., upper inlet). Am I missing something?

You are correct. we cannot only attribute the high variability to the $Hg^0_{gas}$ atmospheric fluctuation. We think that the source of variability is due much lower concentration differences between 20 and 30 cm inlets for both $CO_2$ and $Hg^0_{gas}$ which cause increases variabilities in ratios. We modified the text (page 10, lines 15-17).

Page 12, line 23: "relatively weak and infrequent AMDEs". I quibble but the AMDE reported on Fig. S2 does not look weak to me. About the frequency, how many AMDEs did you observe? You can use a 1.00 ng/m3 threshold to calculate the frequency of occurrence (Angot et al., 2016a; Cobbett et al., 2007; Steffen et al., 2005).

We changed this to infrequent and generally weaker AMDEs. We think it is important to note that the frequency and magnitude of AMDEs indeed is lower than those along the coast. While we see several occasions with air masses with $Hg^0_{gas}$ levels below 1.0 ng m$^{-3}$ (4 to 5 per year), it is very rare to find stronger depletions (e.g., one per year with concentrations <0.5 ng m$^{-3}$). This is dramatically different to studies along the coast where $Hg^0_{gas}$ frequently drops below detection for pronounced time periods.

Page 12, line 31-32: "Total Hg concentrations in all snow samples collected were always much higher than Hgdiss levels". Can we really say that concentrations are "much higher" given the range observed (0-1 ng/L)? By the way, I don't see anywhere Hgtot concentrations from the transect.

Done, we removed "much".

Page 13, line 10: "(. . .) and few studies include inland sites such as Toolik". What is the range of concentrations at these inland sites? Which studies are you referring to?

We clarified that in Douglas & Sturm (2004), snowpack $Hg_{diss}$ concentrations were between 0.5 and 1.7 ng L$^{-1}$ for the interior sites. While our values are even lower, they are not that different from Douglas and Sturm (2004).

Page 14, line 27: "Fresh surface snow". There is no mention of the fact that you collected fresh surface snow in the Materials and Methods Section.

We removed "fresh" and only keep "surface" (i.e., top 3 cm).

Figure 2: Why don't you report the standard deviation for each green bar?

The absence of standard deviation may be due to either narrow differences between the two replicates (i.e., two snow pits), or different snowpack height between the two snow pits (top measurements of Jan. 2015 and Mar. 2016 for example, illustrating the spatial heterogeneity). See Table S2 for more detail about standard deviation values.

Figure 4: What does "soil: organic/mineral" refer to?

This information is now added in the site description in 'Materials and methods' (page 4, lines 8-9).

Figure 7: Please add standard deviations.

As described above, the two samplings were pooled together to constitute a composite sample. Only one sample was analyzed per layer and per site.

Figure 8: I really like this Figure but aren't the various concentrations (<DL, 0.25, 0.50) in the range of the analytical uncertainty?

No, values of 0.25 ng L$^{-1}$ and 0.5 ng L$^{-1}$ are significantly above the detection limit of our method (i.e., above 3 × standard deviation of blank samples).

Figure 9: The colors are really difficult to read. Can you use something else than shades of blue? Maybe a gradient from blue to red.

Done

References

Douglas, T. A. and Sturm, M.: Arctic haze, mercury and the chemical composition of snow across northwestern Alaska, Atmos. Environ., 38(6), 805–820, doi:10.1016/j.atmosenv.2003.10.042, 2004.

Faïn, X., Helmig, D., Hueber, J., Obrist, D. and Williams, M. W.: Mercury dynamics in the Rocky Mountain, Colorado, snowpack, Biogeosciences, 10(6), 3793–3807, doi:10.5194/bg-10-3793-2013, 2013.

Ferrari, C. P., Dommergue, A., Boutron, C. F., Jitaru, P. and Adams, F. C.: Profiles of mercury in the snow pack at Station Nord, Greenland shortly after polar sunrise, Geophys. Res. Lett., 31(3), L03401, doi:10.1029/2003GL018961, 2004.

Obrist, D., Agnan, Y., Jiskra, M., Olson, C. L., Colegrove, D. P., Hueber, J., Moore, C. W., Sonke, J. E. and Helmig, D.: Tundra uptake of atmospheric elemental mercury drives Arctic mercury pollution, Nature, 547(7662), 201–204, doi:10.1038/nature22997, 2017.

Seok, B., Helmig, D., Williams, M. W., Liptzin, D., Chowanski, K. and Hueber, J.: An automated system for continuous measurements of trace gas fluxes through snow: an evaluation of the gas diffusion method at a subalpine forest site, Niwot Ridge, Colorado, Biogeochemistry, 95(1), 95–113, doi:10.1007/s10533-009-9302-3, 2009.

St. Louis, V. L., Sharp, M. J., Steffen, A., May, A., Barker, J., Kirk, J. L., Kelly, D. J. A., Arnott, S. E., Keatley, B. and Smol, J. P.: Some sources and sinks of monomethyl and inorganic mercury on Ellesmere Island in the Canadian high Arctic, Environ. Sci. Technol., 39(8), 2686–2701, doi:10.1021/es049326o, 2005.

**REVIEWER 2**

**General comments**

The manuscript describes a detailed study of Hg in air, snow and soil at an Arctic site, which unlike almost all previous studies is a significant distance from the coast. Given that vast amounts of tundra are inland this study begins to fill in some of the gaps in our knowledge of Hg cycling in these remote regions. Of particular interest are the differences seen between the processes seen at this site when compared to coastal sites. The fact that tundra soils are a sink for atmospheric elemental Hg has important repercussions for future multimedia modelling studies and hints at the potential remobilization of large amounts of Hg from Arctic soils in a warming climate. This should be emphasised more in the Abstract and the Conclusions, in the Abstract particularly the comment on this is brief and hidden in the middle.

Thank you for your comment. We agree with you that the fact that Arctic tundra soils constitute a sink of Hg$^{0}_{gas}$ is an interesting finding for the global understanding of Arctic Hg cycling. This is, however, not the

focus of this paper since we already published that in a previous study (Obrist et al., 2017). We slightly edited both the abstract and conclusion in order to highlight and clarify this point.

The manuscript is however rather long-winded. I think that both the Results and the Discussion section could be shortened significantly, and quite usefully (from the reader's point of view) combined. Just as an example, the discussion of the major ions and the O and H isotope signatures, repeats parts of the pertinent results section. Conversely the results section rather leaves the reader with a sense of 'and what do these results imply?', which is only answered six pages later. I would recommend combing these sections as it will most likely lead naturally to a more concise and less prolix article. If some of the detail in the methods section has already been published perhaps it could be shortened by including more references, if not maybe some of the detail could be moved to the Supplementary material.

We reorganized the manuscript by merging Results and Discussions.

The previous reviewer has comprehensively addressed a number of technical issues, and for me only a few real problems remain.

1. The issue of blowing snow, and where the snow that is being sampled at Toolik comes from.

Indeed, the entire snowpack is subject to important movements that generate uncertainties mentioned by the first reviewer and limiting the observation of trend in the snowpacks. We mention this in regard to vertical Hg concentration patterns, in response to the comments of reviewer 1.

2. The fact that the paper is interesting and adds an important contribution to polar Hg research but is unfortunately not very well presented and at times rather heavy going.

We hope that the key messages are now better presented in this reorganized manuscript version.

3. The importance of atmospheric elemental Hg effectively being sequestered (for the moment) by tundra soils, is not emphasised sufficiently from my point of view.

We previously mentioned the influence of tundra soil in Arctic Hg cycling in Obrist et al., 2017 and highlighted this in the abstract and conclusions.

[revised manuscript text omitted]

Alaska Division of Oil and Gas: Regional geology of the north slope of Alaska, 2008.

Angot, H., Dastoor, A., De Simone, F., Gårdfeldt, K., Gencarelli, C. N., Hedgecock, I. M., Langer, S., Magand, O.,
15   Mastromonaco, M. N., Nordstrøm, C., Pfaffhuber, K. A., Pirrone, N., Ryjkov, A., Selin, N. E., Skov, H., Song, S., Sprovieri, F., Steffen, A., Toyota, K., Travnikov, O., Yang, X. and Dommergue, A.: Chemical cycling and deposition of atmospheric mercury in polar regions: review of recent measurements and comparison with models, Atmospheric Chem. Phys., 16(16), 10735–10763, doi:10.5194/acp-16-10735-2016, 2016a.

Angot, H., Magand, O., Helmig, D., Ricaud, P., Quennehen, B., Gallée, H., Del Guasta, M., Sprovieri, F., Pirrone, N., Savarino,
20   J. and Dommergue, A.: New insights into the atmospheric mercury cycling in central Antarctica and implications on a continental scale, Atmospheric Chem. Phys., 16(13), 8249–8264, doi:10.5194/acp-16-8249-2016, 2016b.

Atwell, L., Hobson, K. A. and Welch, H. E.: Biomagnification and bioaccumulation of mercury in an arctic marine food web: insights from stable nitrogen isotope analysis, Can. J. Fish. Aquat. Sci., 55(5), 1114–1121, doi:10.1139/f98-001, 1998.

Barker, A. J., Douglas, T. A., Jacobson, A. D., McClelland, J. W., Ilgen, A. G., Khosh, M. S., Lehn, G. O. and Trainor, T. P.:
25   Late season mobilization of trace metals in two small Alaskan arctic watersheds as a proxy for landscape scale permafrost active layer dynamics, Chem. Geol., 381, 180–193, doi:10.1016/j.chemgeo.2014.05.012, 2014.

Bergin, M. H., Jaffrezo, J.-L., Davidson, C. I., Dibb, J. E., Pandis, S. N., Hillamo, R., Maenhaut, W., Kuhns, H. D. and Makela, T.: The contributions of snow, fog, and dry deposition to the summer flux of anions and cations at Summit, Greenland, J. Geophys. Res. Atmospheres, 100(D8), 16275–16288, doi:10.1029/95JD01267, 1995.

30   Brooks, S., Lindberg, S., Southworth, G. and Arimoto, R.: Springtime atmospheric mercury speciation in the McMurdo, Antarctica coastal region, Atmos. Environ., 42(12), 2885–2893, doi:10.1016/j.atmosenv.2007.06.038, 2008.

Brooks, S. B., Saiz-Lopez, A., Skov, H., Lindberg, S. E., Plane, J. M. C. and Goodsite, M. E.: The mass balance of mercury in the springtime arctic environment, Geophys. Res. Lett., 33(L13812), doi:10.1029/2005GL025525, 2006.

de Caritat, P., Hall, G., Gìslason, S., Belsey, W., Braun, M., Goloubeva, N. I., Olsen, H. K., Scheie, J. O. and Vaive, J. E.: Chemical composition of arctic snow: concentration levels and regional distribution of major elements, Sci. Total Environ., 336(1), 183–199, doi:10.1016/j.scitotenv.2004.05.031, 2005.

Cherry, J. E., Déry, S. J., Cheng, Y., Stieglitz, M., Jacobs, A. S. and Pan, F.: Climate and hydrometeorology of the Toolik Lake region and the Kuparuk River basin, in Alaska's changing arctic: ecological consequences for tundra, streams, and lakes, edited by J. E. Hobbie and G. W. Kling, pp. 21–60, Oxford University Press, New York., 2014.

Cobbett, F. D., Steffen, A., Lawson, G. and van Heyst, B. J.: GEM fluxes and atmospheric mercury concentrations (GEM, RGM and Hgp) in the Canadian Arctic at Alert, Nunavut, Canada (February–June 2005), Atmos. Environ., 41(31), 6527–6543, doi:10.1016/j.atmosenv.2007.04.033, 2007.

Corbitt, E. S., Jacob, D. J., Holmes, C. D., Streets, D. G. and Sunderland, E. M.: Global source-receptor relationships for mercury deposition under present-day and 2050 emissions scenarios, Environ. Sci. Technol., 45(24), 10477–10484, doi:10.1021/es202496y, 2011.

Craig, H.: Isotopic variations in meteoric waters, Science, 133(3465), 1702–1703, doi:10.1126/science.133.3465.1702, 1961.

Dominé, F. and Shepson, P. B.: Air-snow interactions and atmospheric chemistry, Science, 297(5586), 1506–1510, doi:10.1126/science.1074610, 2002.

Dommergue, A., Ferrari, C. P., Poissant, L., Gauchard, P.-A. and Boutron, C. F.: Diurnal cycles of gaseous mercury within the snowpack at Kuujjuarapik/Whapmagoostui, Québec, Canada, Environ. Sci. Technol., 37(15), 3289–3297, doi:10.1021/es026242b, 2003.

Dommergue, A., Sprovieri, F., Pirrone, N., Ebinghaus, R., Brooks, S., Courteaud, J. and Ferrari, C. P.: Overview of mercury measurements in the Antarctic troposphere, Atmospheric Chem. Phys., 10(7), 3309–3319, doi:10.5194/acp-10-3309-2010, 2010.

Douglas, T. A. and Sturm, M.: Arctic haze, mercury and the chemical composition of snow across northwestern Alaska, Atmos. Environ., 38(6), 805–820, doi:10.1016/j.atmosenv.2003.10.042, 2004.

Douglas, T. A., Sturm, M., Simpson, W. R., Brooks, S., Lindberg, S. E. and Perovich, D. K.: Elevated mercury measured in snow and frost flowers near Arctic sea ice leads, Geophys. Res. Lett., 32(4), L04502, doi:10.1029/2004GL022132, 2005.

Douglas, T. A., Sturm, M., Simpson, W. R., Blum, J. D., Alvarez-Aviles, L., Keeler, G. J., Perovich, D. K., Biswas, A. and Johnson, K.: Influence of snow and ice crystal formation and accumulation on mercury deposition to the Arctic, Environ. Sci. Technol., 42(5), 1542–1551, doi:10.1021/es070502d, 2008.

Douglas, T. A., Loseto, L. L., Macdonald, R. W., Outridge, P., Dommergue, A., Poulain, A., Amyot, M., Barkay, T., Berg, T., Chételat, J., Constant, P., Evans, M., Ferrari, C., Gantner, N., Johnson, M. S., Kirk, J., Kroer, N., Larose, C., Lean, D., Nielsen, T. G., Poissant, L., Rognerud, S., Skov, H., Sørensen, S., Wang, F., Wilson, S. and Zdanowicz, C. M.: The fate of mercury in arctic terrestrial and aquatic ecosystems, a review, Environ. Chem., 9(4), 321–355, doi:10.1071/EN11140, 2012.

Douglas, T. A., Sturm, M., Blum, J. D., Polashenski, C., Stuefer, S., Hiemstra, C., Steffen, A., Filhol, S. and Prevost, R.: A pulse of mercury and major ions in snowmelt runoff from a small arctic Alaska watershed, Environ. Sci. Technol., 51(19), 11145–11155, doi:10.1021/acs.est.7b03683, 2017.

Driscoll, C. T., Mason, R. P., Chan, H. M., Jacob, D. J. and Pirrone, N.: Mercury as a global pollutant: sources, pathways, and effects, Environ. Sci. Technol., 47(10), 4967–4983, doi:10.1021/es305071v, 2013.

Enrico, M., Le Roux, G., Heimbürger, L.-E., Van Beek, P., Souhaut, M., Chmeleff, J. and Sonke, J. E.: Holocene atmospheric mercury levels reconstructed from peat bog mercury stable isotopes, Environ. Sci. Technol., 51(11), 5899–5906, doi:10.1021/acs.est.6b05804, 2017.

Essery, R. and Pomeroy, J.: Vegetation and topographic control of wind-blown snow distributions in distributed and aggregated simulations for an arctic tundra basin, J. Hydrometeorol., 5(5), 735–744, doi:10.1175/1525-7541(2004)005<0735:VATCOW>2.0.CO;2, 2004.

Essery, R., Li, L. and Pomeroy, J.: A distributed model of blowing snow over complex terrain, Hydrol. Process., 13(1415), 2423–2438, doi:10.1002/(SICI)1099-1085(199910)13:14/15<2423::AID-HYP853>3.0.CO;2-U, 1999.

Faïn, X., Grangeon, S., Bahlmann, E., Fritsche, J., Obrist, D., Dommergue, A., Ferrari, C. P., Cairns, W., Ebinghaus, R., Barbante, C., Cescon, P. and Boutron, C.: Diurnal production of gaseous mercury in the alpine snowpack before snowmelt, J. Geophys. Res., 112(D21311), doi:10.1029/2007JD008520, 2007.

Faïn, X., Ferrari, C. P., Dommergue, A., Albert, M., Battle, M., Arnaud, L., Barnola, J.-M., Cairns, W., Barbante, C. and Boutron, C.: Mercury in the snow and firn at Summit Station, Central Greenland, and implications for the study of past atmospheric mercury levels, Atmos Chem Phys, 8(13), 3441–3457, doi:10.5194/acp-8-3441-2008, 2008.

Faïn, X., Obrist, D., Pierce, A., Barth, C., Gustin, M. S. and Boyle, D. P.: Whole-watershed mercury balance at Sagehen Creek, Sierra Nevada, CA, Geochim. Cosmochim. Acta, 75(9), 2379–2392, doi:10.1016/j.gca.2011.01.041, 2011.

Faïn, X., Helmig, D., Hueber, J., Obrist, D. and Williams, M. W.: Mercury dynamics in the Rocky Mountain, Colorado, snowpack, Biogeosciences, 10(6), 3793–3807, doi:10.5194/bg-10-3793-2013, 2013.

Ferrari, C. P., Dommergue, A., Boutron, C. F., Jitaru, P. and Adams, F. C.: Profiles of mercury in the snow pack at Station Nord, Greenland shortly after polar sunrise, Geophys. Res. Lett., 31(3), L03401, doi:10.1029/2003GL018961, 2004.

Ferrari, C. P., Gauchard, P.-A., Aspmo, K., Dommergue, A., Magand, O., Bahlmann, E., Nagorski, S., Temme, C., Ebinghaus, R., Steffen, A., Banic, C., Berg, T., Planchon, F., Barbante, C., Cescon, P. and Boutron, C. F.: Snow-to-air exchanges of mercury in an Arctic seasonal snow pack in Ny-Ålesund, Svalbard, Atmos. Environ., 39(39), 7633–7645, doi:10.1016/j.atmosenv.2005.06.058, 2005.

Ferrari, C. P., Padova, C., Faïn, X., Gauchard, P.-A., Dommergue, A., Aspmo, K., Berg, T., Cairns, W., Barbante, C., Cescon, P., Kaleschke, L., Richter, A., Wittrock, F. and Boutron, C.: Atmospheric mercury depletion event study in Ny-Ålesund (Svalbard) in spring 2005. Deposition and transformation of Hg in surface snow during springtime, Sci. Total Environ., 397(1–3), 167–177, doi:10.1016/j.scitotenv.2008.01.064, 2008.

Fitzgerald, W. F., Engstrom, D. R., Lamborg, C. H., Tseng, C.-M., Balcom, P. H. and Hammerschmidt, C. R.: Modern and historic atmospheric mercury fluxes in Northern Alaska: global sources and arctic depletion, Environ. Sci. Technol., 39(2), 557–568, doi:10.1021/es049128x, 2005.

Fitzgerald, W. F., Hammerschmidt, C. R., Engstrom, D. R., Balcom, P. H., Lamborg, C. H. and Tseng, C.-M.: Mercury in the Alaskan arctic, in Alaska's changing arctic: ecological consequences for tundra, streams, and lakes, edited by J. E. Hobbie and G. W. Kling, pp. 287–302, Oxford University Press, New York., 2014.

Garbarino, J. R., Snyder-Conn, E., Leiker, T. J. and Hoffman, G. L.: Contaminants in Arctic snow collected over Northwest Alaskan sea ice, Water. Air. Soil Pollut., 139(1–4), 183–214, doi:10.1023/A:1015808008298, 2002.

Gat, J. R.: Isotope hydrology: a study of the water cycle, World Scientific, London., 2010.

King, M. D. and Simpson, W. R.: Extinction of UV radiation in arctic snow at Alert, Canada (82°N), J. Geophys. Res. Atmospheres, 106(D12), 12499–12507, doi:10.1029/2001JD900006, 2001.

Kirk, J. L., St. Louis, V. L. and Sharp, M. J.: Rapid reduction and reemission of mercury deposited into snowpacks during atmospheric mercury depletion events at Churchill, Manitoba, Canada, Environ. Sci. Technol., 40(24), 7590–7596, doi:10.1021/es061299+, 2006.

Krnavek, L., Simpson, W. R., Carlson, D., Domine, F., Douglas, T. A. and Sturm, M.: The chemical composition of surface snow in the Arctic: Examining marine, terrestrial, and atmospheric influences, Atmos. Environ., 50(Supplement C), 349–359, doi:10.1016/j.atmosenv.2011.11.033, 2012.

Lalonde, J. D., Poulain, A. J. and Amyot, M.: The role of mercury redox reactions in snow on snow-to-air mercury transfer, Environ. Sci. Technol., 36(2), 174–178, doi:10.1021/es010786g, 2002.

Landers, D. H., Ford, J., Gubala, C., Monetti, M., Lasorsa, B. K. and Martinson, J.: Mercury in vegetation and lake sediments from the U.S. Arctic, Water. Air. Soil Pollut., 80(1–4), 591–601, doi:10.1007/BF01189711, 1995.

Lindberg, S. E., Hanson, P. J., Meyers, T. P. and Kim, K.-H.: Air/surface exchange of mercury vapor over forests—the need for a reassessment of continental biogenic emissions, Atmos. Environ., 32(5), 895–908, doi:10.1016/S1352-2310(97)00173-8, 1998.

Liptzin, D., Williams, M. W., Helmig, D., Seok, B., Filippa, G., Chowanski, K. and Hueber, J.: Process-level controls on $CO_2$ fluxes from a seasonally snow-covered subalpine meadow soil, Niwot Ridge, Colorado, Biogeochemistry, 95(1), 151–166, doi:10.1007/s10533-009-9303-2, 2009.

Mann, E., Meyer, T., Mitchell, C. P. J. and Wania, F.: Mercury fate in ageing and melting snow: development and testing of a controlled laboratory system, J. Environ. Monit., 13(10), 2695–2702, doi:10.1039/C1EM10297D, 2011.

Mann, E., Ziegler, S., Mallory, M. and O'Driscoll, N.: Mercury photochemistry in snow and implications for arctic ecosystems, Environ. Rev., 22(4), 331–345, doi:10.1139/er-2014-0006, 2014.

Mann, E. A., Mallory, M. L., Ziegler, S. E., Tordon, R. and O'Driscoll, N. J.: Mercury in Arctic snow: quantifying the kinetics of photochemical oxidation and reduction, Sci. Total Environ., 509–510, 115–132, doi:10.1016/j.scitotenv.2014.07.056, 2015.

Monson, R. K., Burns, S. P., Williams, M. W., Delany, A. C., Weintraub, M. and Lipson, D. A.: The contribution of beneath-snow soil respiration to total ecosystem respiration in a high-elevation, subalpine forest, Glob. Biogeochem. Cycles, 20(GB3030), doi:10.1029/2005GB002684, 2006.

Moore, C. W., Obrist, D., Steffen, A., Staebler, R. M., Douglas, T. A., Richter, A. and Nghiem, S. V.: Convective forcing of mercury and ozone in the Arctic boundary layer induced by leads in sea ice, Nature, 506(7486), 81–84, doi:10.1038/nature12924, 2014.

National Atmospheric Deposition Program: (NRSP-3), NADP Program Office, Illinois State Water Survey, University of Illinois, Champaign, IL 61820., 2017.

Nerentorp Mastromonaco, M., Gårdfeldt, K., Jourdain, B., Abrahamsson, K., Granfors, A., Ahnoff, M., Dommergue, A., Méjean, G. and Jacobi, H.-W.: Antarctic winter mercury and ozone depletion events over sea ice, Atmos. Environ., 129, 125–132, doi:10.1016/j.atmosenv.2016.01.023, 2016.

Norman, A. L., Barrie, L. A., Toom-Sauntry, D., Sirois, A., Krouse, H. R., Li, S. M. and Sharma, S.: Sources of aerosol sulphate at Alert: apportionment using stable isotopes, J. Geophys. Res. Atmospheres, 104(D9), 11619–11631, doi:10.1029/1999JD900078, 1999.

Obrist, D., Tas, E., Peleg, M., Matveev, V., Faïn, X., Asaf, D. and Luria, M.: Bromine-induced oxidation of mercury in the mid-latitude atmosphere, Nat. Geosci., 4(1), 22–26, doi:10.1038/ngeo1018, 2011.

Obrist, D., Pokharel, A. K. and Moore, C.: Vertical profile measurements of soil air suggest immobilization of gaseous elemental mercury in mineral soil, Environ. Sci. Technol., 48(4), 2242–2252, doi:10.1021/es4048297, 2014.

Obrist, D., Agnan, Y., Jiskra, M., Olson, C. L., Colegrove, D. P., Hueber, J., Moore, C. W., Sonke, J. E. and Helmig, D.: Tundra uptake of atmospheric elemental mercury drives Arctic mercury pollution, Nature, 547(7662), 201–204, doi:10.1038/nature22997, 2017.

Oechel, W. C., Vourlitis, G. and Hastings, S. J.: Cold season $CO_2$ emission from arctic soils, Glob. Biogeochem. Cycles, 11(2), 163–172, doi:10.1029/96GB03035, 1997.

Pearson, C., Schumer, R., Trustman, B. D., Rittger, K., Johnson, D. W. and Obrist, D.: Nutrient and mercury deposition and storage in an alpine snowpack of the Sierra Nevada, USA, Biogeosciences, 12(12), 3665–3680, doi:10.5194/bg-12-3665-2015, 2015.

Poulain, A. J., Lalonde, J. D., Amyot, M., Shead, J. A., Raofie, F. and Ariya, P. A.: Redox transformations of mercury in an Arctic snowpack at springtime, Atmos. Environ., 38(39), 6763–6774, doi:10.1016/j.atmosenv.2004.09.013, 2004.

Schroeder, W. H. and Munthe, J.: Atmospheric mercury—An overview, Atmos. Environ., 32(5), 809–822, doi:10.1016/S1352-2310(97)00293-8, 1998.

Schroeder, W. H., Anlauf, K. G., Barrie, L. A., Lu, J. Y., Steffen, A., Schneeberger, D. R. and Berg, T.: Arctic springtime depletion of mercury, Nature, 394, 331–332, doi:10.1038/28530, 1998.

Selin, N. E.: Global biogeochemical cycling of mercury: a review, Annu. Rev. Environ. Resour., 34(1), 43–63, doi:10.1146/annurev.environ.051308.084314, 2009.

Seok, B., Helmig, D., Williams, M. W., Liptzin, D., Chowanski, K. and Hueber, J.: An automated system for continuous measurements of trace gas fluxes through snow: an evaluation of the gas diffusion method at a subalpine forest site, Niwot Ridge, Colorado, Biogeochemistry, 95(1), 95–113, doi:10.1007/s10533-009-9302-3, 2009.

Shaver, G. R. and Chapin, F. S.: Production: biomass relationships and element cycling in contrasting arctic vegetation types, Ecol. Monogr., 61(1), 1–31, doi:10.2307/1942997, 1991.

Siegenthaler, U. and Oeschger, H.: Correlation of [18]O in precipitation with temperature and altitude, Nature, 285(5763), 314–317, doi:10.1038/285314a0, 1980.

Simpson, W. R., von Glasow, R., Riedel, K., Anderson, P., Ariya, P., Bottenheim, J., Burrows, J., Carpenter, L. J., Frieß, U., Goodsite, M. E., Heard, D., Hutterli, M., Jacobi, H.-W., Kaleschke, L., Neff, B., Plane, J., Platt, U., Richter, A., Roscoe, H.,

Sander, R., Shepson, P., Sodeau, J., Steffen, A., Wagner, T. and Wolff, E.: Halogens and their role in polar boundary-layer ozone depletion, Atmos Chem Phys, 7(16), 4375–4418, doi:10.5194/acp-7-4375-2007, 2007.

Snyder-Conn, E., Garbarino, J. R., Hoffman, G. L. and Oelkers, A.: Soluble trace elements and total mercury in arctic alaskan snow, Arctic, 50(3), 201–215, 1997.

5  Sprovieri, F., Pirrone, N., Ebinghaus, R., Kock, H. and Dommergue, A.: A review of worldwide atmospheric mercury measurements, Atmospheric Chem. Phys., 10(17), 8245–8265, doi:10.5194/acp-10-8245-2010, 2010.

St. Louis, V. L., Sharp, M. J., Steffen, A., May, A., Barker, J., Kirk, J. L., Kelly, D. J. A., Arnott, S. E., Keatley, B. and Smol, J. P.: Some sources and sinks of monomethyl and inorganic mercury on Ellesmere Island in the Canadian high Arctic, Environ. Sci. Technol., 39(8), 2686–2701, doi:10.1021/es049326o, 2005.

10  Steffen, A., Schroeder, W., Bottenheim, J., Narayan, J. and Fuentes, J. D.: Atmospheric mercury concentrations: measurements and profiles near snow and ice surfaces in the Canadian Arctic during Alert 2000, Atmos. Environ., 36(15–16), 2653–2661, doi:10.1016/S1352-2310(02)00112-7, 2002.

Steffen, A., Douglas, T., Amyot, M., Ariya, P., Aspmo, K., Berg, T., Bottenheim, J., Brooks, S., Cobbett, F., Dastoor, A., Dommergue, A., Ebinghaus, R., Ferrari, C., Gardfeldt, K., Goodsite, M. E., Lean, D., Poulain, A. J., Scherz, C., Skov, H.,
15  Sommar, J. and Temme, C.: A synthesis of atmospheric mercury depletion event chemistry in the atmosphere and snow, Atmos Chem Phys, 8(6), 1445–1482, doi:10.5194/acp-8-1445-2008, 2008.

Steffen, A., Bottenheim, J., Cole, A., Douglas, T. A., Ebinghaus, R., Friess, U., Netcheva, S., Nghiem, S., Sihler, H. and Staebler, R.: Atmospheric mercury over sea ice during the OASIS-2009 campaign, Atmospheric Chem. Phys., 13(14), 7007–7021, doi:10.5194/acp-13-7007-2013, 2013.

20  Steffen, A., Bottenheim, J., Cole, A., Ebinghaus, R., Lawson, G. and Leaitch, W. R.: Atmospheric mercury speciation and mercury in snow over time at Alert, Canada, Atmos Chem Phys, 14(5), 2219–2231, doi:10.5194/acp-14-2219-2014, 2014.

Sturm, M. and Liston, G. E.: The snow cover on lakes of the Arctic Coastal Plain of Alaska, U.S.A., J. Glaciol., 49(166), 370–380, doi:10.3189/172756503781830539, 2003.

Toom-Sauntry, D. and Barrie, L. A.: Chemical composition of snowfall in the high Arctic: 1990–1994, Atmos. Environ.,
25  36(15–16), 2683–2693, doi:10.1016/S1352-2310(02)00115-2, 2002.

Uematsu, M., Kinoshita, K. and Nojiri, Y.: Scavenging of insoluble particles from the marine atmosphere over the sub-arctic north Pacific, J. Atmospheric Chem., 35(2), 151–163, doi:10.1023/A:1006219028497, 2000.

US EPA: Method 1631: Mercury in water by oxidation, purge and trap, and cold vapor atomic fluorescence spectrometry, United States Environmental Protection Agency., 2002.

30  Van Dam, B., Helmig, D., Burkhart, J. F., Obrist, D. and Oltmans, S. J.: Springtime boundary layer $O_3$ and GEM depletion at Toolik Lake, Alaska, J. Geophys. Res. Atmospheres, 118(8), 3382–3391, doi:10.1002/jgrd.50213, 2013.

[revised manuscript text omitted]

**Supplementary material:**

**Table S1: Geographical coordinates of transect sampling sites.**

| site | latitude | longitude | elevation (m) | distance to Dalton Highway (m) |
|------|----------|-----------|---------------|-------------------------------|
| transect 1 | 68.7605° N | 148.8659° W | 501 | 730 |
| transect 2 | 69.0350° N | 148.8258° W | 400 | 230 |
| transect 3 | 69.4212° N | 148.6691° W | 333 | 210 |
| transect 4 | 69.5692° N | 148.6049° W | 145 | 560 |
| transect 5 | 69.6741° N | 148.7003° W | 118 | 240 |
| transect 6 | 69.8324° N | 148.7555° W | 84 | 140 |
| transect 7 | 70.0031° N | 148.6804° W | 46 | 250 |
| transect 8 | 70.1323° N | 148.4896° W | 21 | 250 |

**Table S2: Summary total (Hg$_{tot}$) and dissolved Hg (Hg$_{diss}$) concentrations in snowpack and surface snow layers on the tundra and lake at Toolik Field Station.**

| date | location | height (cm) | Hg$_{tot}$ (ng L$^{-1}$) | | Hg$_{diss}$ (ng L$^{-1}$) | |
|---|---|---|---|---|---|---|
| | | | mean | SD[a] | mean | SD[a] |
| Oct. 14th 2014 | tundra | 36 | 0.22 | 0.04 | 0.18 | 0.00 |
| | | 25 | 0.24 | 0.04 | 0.14 | 0.08 |
| | | 13 | 0.17 | 0.18 | 0.09 | 0.08 |
| Dec. 7th 2014 | tundra | surface | 0.21 | 0.03 | <DL[b] | – |
| | | 21 | 0.43 | 0.55 | 0.12 | 0.03 |
| | | 11 | 1.06 | 1.30 | 0.16 | 0.06 |
| Dec. 31st 2014 | tundra | surface | 0.36 | 0.28 | 0.17 | 0.06 |
| Jan. 26th 2015 | tundra | surface | 0.27 | 0.16 | 0.12 | 0.05 |
| | | 50 | 0.36 | – | 0.26 | – |
| | | 37 | 6.23 | – | <DL[b] | – |
| | | 25 | 0.46 | 0.16 | 0.11 | 0.11 |
| | | 10 | 0.28 | 0.17 | 0.08 | 0.06 |
| | lake | 8 | 0.31 | 0.06 | 0.21 | 0.05 |
| Feb. 20th 2015 | tundra | surface | 0.18 | 0.03 | 0.13 | 0.02 |
| Mar. 8th 2015 | tundra | 48 | 0.66 | 0.10 | 0.15 | 0.02 |
| | | 37 | 0.59 | 0.36 | 0.11 | 0.10 |
| | | 25 | 0.19 | 0.03 | 0.08 | 0.05 |
| | | 13 | 0.29 | 0.05 | <DL[b] | – |
| | lake | 11 | 0.74 | 0.02 | <DL[b] | – |
| Apr. 3rd 2015 | tundra | surface | – | – | 0.24 | 0.06 |
| Apr. 17th 2015 | tundra | surface | 0.87 | 0.25 | 0.21 | 0.04 |
| | | 35 | 1.03 | 0.45 | 0.28 | 0.01 |
| | | 25 | 0.61 | 0.05 | 0.31 | 0.08 |
| | | 15 | 1.24 | 0.41 | 0.29 | 0.15 |
| | | 5 | 0.91 | 0.62 | 0.33 | 0.22 |
| | lake | 11 | 1.43 | 0.23 | 0.09 | 0.07 |

[a] standard deviation (n = 2 for snowpack and n = 3 for surface snow)

[b] below detection limit

5    "–" no data

| date | location | height (cm) | Hg$_{tot}$ (ng L$^{-1}$) | | Hg$_{diss}$ (ng L$^{-1}$) | |
|---|---|---|---|---|---|---|
| | | | mean | SD[a] | mean | SD[a] |
| Oct. 19th 2015 | tundra | surface | 0.32 | 0.03 | 0.12 | 0.07 |
| Nov. 15th 2015 | tundra | surface | 0.46 | 0.17 | 0.26 | 0.13 |
| Dec. 5th 2015 | tundra | surface | 0.19 | 0.01 | 0.10 | 0.09 |
| | | 34 | 0.32 | – | 0.12 | – |
| | | 26 | 0.78 | – | 0.23 | – |
| | | 16 | 0.43 | 0.12 | 0.20 | 0.04 |
| | | 8 | 0.37 | 0.08 | 0.27 | 0.13 |
| Jan. 13th 2016 | tundra | surface | 0.82 | 0.54 | 0.13 | 0.08 |
| Jan. 29th 2016 | tundra | surface | 0.38 | 0.15 | 0.29 | 0.02 |
| | | 35 | 1.00 | – | 0.18 | – |
| | | 25 | 0.69 | 0.75 | 0.16 | 0.05 |
| | | 15 | 0.56 | 0.18 | 0.19 | 0.05 |
| | | 5 | 0.64 | – | 0.17 | – |
| | lake | 5 | 1.63 | 0.63 | 0.24 | 0.06 |
| Feb. 20th 2016 | tundra | surface | 0.60 | 0.15 | 0.24 | 0.08 |
| Mar. 3rd 2016 | tundra | surface | 0.41 | 0.04 | 0.33 | 0.05 |
| Mar. 25th 2016 | tundra | surface | 1.01 | 0.07 | 0.49 | 0.03 |
| | | 50 | 0.58 | – | 0.13 | – |
| | | 40 | 0.85 | – | 0.14 | – |
| | | 30 | 0.54 | – | 0.10 | – |
| | | 20 | 0.31 | 0.06 | <DL[b] | – |
| | | 10 | 0.46 | 0.13 | 0.19 | 0.09 |
| | lake | 13 | 0.41 | 0.16 | 0.17 | 0.02 |
| Apr. 2nd 2016 | tundra | surface | 1.46 | 0.16 | 1.15 | 0.15 |
| Apr. 13th 2016 | tundra | surface | 0.21 | 0.03 | 0.13 | 0.01 |
| May 1st 2016 | tundra | surface | 0.66 | 0.23 | 0.23 | 0.08 |
| | | 33 | 0.21 | 0.12 | 0.12 | 0.11 |
| | | 23 | 1.07 | 1.20 | 0.13 | 0.06 |
| | | 15 | 0.43 | – | 0.13 | – |
| | lake | 13 | 0.24 | 0.07 | 0.12 | 0.11 |

[a] standard deviation (n = 2 for snowpack and n = 3 for surface snow)
[b] below detection limit
"–" no data

**Table S3: Summary total (Hg$_{tot}$) and dissolved Hg (Hg$_{diss}$) concentrations in snowpack on the tundra of transect sampling sites.**

| site | height (cm) | Hg$_{tot}$ (ng L$^{-1}$) | Hg$_{diss}$ (ng L$^{-1}$) |
|---|---|---|---|
| | 40 | 0.22 | 0.13 |
| transect 1 | 30 | 0.22 | 0.11 |
| | 20 | 1.32 | 0.17 |
| | 33 | 0.27 | 0.13 |
| transect 2 | 23 | 0.15 | <DL[a] |
| | 13 | 1.79 | 0.28 |
| | 45 | 0.25 | 0.13 |
| | 35 | 0.10 | <DL[a] |
| transect 3 | 25 | 0.18 | <DL[a] |
| | 15 | 0.18 | <DL[a] |
| | 32 | 0.57 | 0.38 |
| transect 4 | 22 | 0.38 | 0.13 |
| | 12 | 0.34 | 0.24 |
| | 40 | 0.60 | 0.34 |
| | 30 | 0.32 | 0.23 |
| transect 5 | 20 | 0.36 | 0.14 |
| | 10 | 0.92 | 0.23 |
| | 61 | 0.86 | 0.46 |
| transect 6 | 51 | 1.47 | 0.86 |
| | 41 | 0.54 | 0.19 |
| transect 7 | 29 | 0.42 | 0.29 |
| | 19 | 0.78 | 0.26 |
| transect 8 | 25 | 3.79 | 0.74 |
| | 15 | 0.73 | 0.20 |

[a] below detection limit

[Figure]

**Figure S1: Snow tower installation over the arctic tundra at Toolik Field Station.**

[Figure]

**Figure S2**: $Hg^0_{gas}$ concentrations (3-hours averages) in the atmosphere and snowpack interstitial air (10 and 20 cm above the ground surface), as well as $Hg^{II}$ and $O_3$ atmospheric measurements during a week in spring 2016, including an AMDE, measured at Toolik Field Station. The snow height was between 25 and 28 cm and surface snow $Hg_{tot}$ concentrations between $1.00 \pm 0.07$ (March $25^{th}$) and $1.46 \pm 0.16$ ng $L^{-1}$ (April $2^{nd}$). The gray bars indicate nighttime periods.

[revised manuscript text omitted]

**4 Discussions**

**4.1 Gas-phase mercury exchanges in the snowpack and photochemical processes**

We continuously measured $Hg^\theta_{gas}$ concentrations and diffusion patterns in the atmosphere snowpack soil continuum throughout the two snow seasons at Toolik (Fig. 3 and Fig. 4). The measurement of trace gas concentration patterns allows determination of the direction of atmosphere surface exchanges as trace gas exchange must follow concentration gradients (Sommerfeld et al., 1996). In a previous paper, we reported a small rate of continuous $Hg^\theta_{gas}$ deposition from the atmosphere to the tundra—measured by a micrometeorological tower—during much of the snow-covered season, with the exception of short time periods in spring during the occurrence of AMDEs at Toolik (Obrist et al., 2017). Here, we show that these flux measurements are supported by consistent $Hg^\theta_{gas}$ concentration gradients that existed through both seasons and that showed that snowpack $Hg^\theta_{gas}$ concentrations were consistently lower than atmospheric levels in the snowpack. In addition, snowpack $Hg^\theta_{gas}$ declined with depth in the snowpack and were lowest in the underlying soil, showing evidence of a consistent $Hg^\theta_{gas}$ concentration gradient from the atmosphere to surface snow to tundra soils.

It is important to mention that our $Hg^\theta_{gas}$ concentration profiles in the arctic snowpack are inherently different to patterns observed in lower latitude snowpacks. In the Rocky Mountains, for example, the upper snowpack showed strong enrichments of $Hg^\theta_{gas}$ throughout most of the winter (i.e., up to 6 times higher concentrations than in the atmosphere; Fig. 3b, Faïn et al., 2013). Such $Hg^\theta_{gas}$ concentration enrichments were attributed to strong photochemically-initiated reduction of snow-bound $Hg^{II}$ to $Hg^\theta_{gas}$ (Lalonde et al., 2002). The implications of $Hg^\theta_{gas}$ production is that subsequent volatilization of the $Hg^\theta_{gas}$ from

the porous snowpack to the atmosphere can alleviate atmospheric deposition loads and it is estimated that globally 50% of snow-bound Hg is volatilized back to the atmosphere prior to snowmelt (Corbitt et al., 2011). Our trace gas concentration measurements showed that $Hg^{\theta}_{gas}$ re-volatilization is missing in this interior tundra snowpack during most of the winter, and the absence of direct solar radiation likely explains the lack of photochemical $Hg^{\theta}_{gas}$ formation and volatilization between December through mid-January. Yet, springtime is a photochemically active period in the arctic when strong $Hg^{\theta}_{gas}$ volatilization from snow has been reported further north along the Arctic Ocean coast (Brooks et al., 2006; Kirk et al., 2006). Even in late spring, when abundant solar radiation is present, however, $Hg^{\theta}_{gas}$ volatilization losses were rare and largely limited to periods of active AMDEs. We speculate that a reason for the general lack of $Hg^{\theta}_{gas}$ formation and volatilization in snow includes substrate limitation due to very low total snow Hg concentrations (Fig. 2, see below), several times lower compared to concentrations in temperate snowpacks (Faïn et al., 2013).

A key question pertaining to the wintertime $Hg^{\theta}_{gas}$ concentration profiles and measured deposition is if the observed $Hg^{\theta}_{gas}$ deposition and concentration declines in the snowpack are driven by $Hg^{\theta}_{gas}$ sinks in the snowpack or by $Hg^{\theta}_{gas}$ uptake by the underlying tundra. Sinks of $Hg^{\theta}_{gas}$ in the snowpack have been observed in a few studies (Dommergue et al., 2003; Faïn et al., 2008, 2013) and attributed to dark oxidation of $Hg^{\theta}_{gas}$ to divalent, non-volatile $Hg^{II}$, possibly including oxidation by halogen species, $O_3$, or related to NOx chemistry. To address this question, $CO_2$ serves as a tracer for soil contributions since soils are the only active wintertime sources of $CO_2$. Because the concentration gradients are directly related to the flux ratios, the consistent $\Delta_{Hg^{\theta}_{gas}}/\Delta_{CO_2}$ values through the snowpack provide evidence that the two trace gases are transported similarly through the snowpack, although in opposite direction as shown by negative ratio values (Fig. 5). This illustrates that the $Hg^{\theta}_{gas}$ uptake occurs in soils rather than in the snowpack.

A soil $Hg^{\theta}_{gas}$ sink was previously reported to occur in temperate soils (Obrist et al., 2014), although the mechanisms for the $Hg^{\theta}_{gas}$ sinks are currently not clear. Our observations hence indicate that the tundra snowpack was not an active sink for atmospheric $Hg^{\theta}_{gas}$, but rather represented a porous and relatively unreactive matrix that reflected a strong concentration gradient between the atmosphere and tundra soils. The wintertime atmosphere–snowpack–soil $Hg^{\theta}_{gas}$ concentration profiles at Toolik were consistent with a measured net deposition of $Hg^{\theta}_{gas}$ throughout winter (Fig. 2 and Fig. 4; Obrist et al., 2017). Both net flux measurements and observations within the snowpack hence suggest that a soil $Hg^{\theta}_{gas}$ sink was active throughout the Arctic winter, notably under very cold wintertime soil temperatures as low as −15 °C. It is notable that $\Delta_{Hg^{\theta}_{gas}}/\Delta_{CO_2}$ ratios in the upper snowpack (i.e., between 20 and 30 cm height) were more variable compared to lower snowpack heights, which we attribute to higher variability in upper snowpack concentrations due to variable atmospheric $Hg^{\theta}_{gas}$ levels. Strong variations of $Hg^{\theta}_{gas}$ concentrations were observed in the atmosphere—particularly during springtime (Fig. S2)—, supporting the notion of snowpack as a highly porous matrix that is in strong diffusive and advective exchange with atmospheric trace gas concentrations (Fig. 5).

Springtime was the only period when occasional $Hg^{\theta}_{gas}$ concentration enhancements in the uppermost snowpack were present (less than 5% of the time) while deeper snowpack $Hg^{\theta}_{gas}$ concentrations remained at low levels (Fig. S2). Our measurements

of $Hg^0_{gas}$ showed that during March and April was the only time when we observed small rates of $Hg^0_{gas}$ formation in the uppermost snowpack layer, suggesting some photochemical reduction and re-volatilization of $Hg^0_{gas}$ after AMDE Hg deposition. However, the $Hg^0_{gas}$ production was small, limited in time, and no photochemical $Hg^0_{gas}$ production and re-emission was observed from deeper snow layers suggesting that the process was limited to the snowpack surface. March and April were also the only months when we observed periods of net $Hg^0_{gas}$ flux emission from the tundra ecosystem to the atmosphere (Obrist et al., 2017), in further support of the typical Hg dynamics reported during AMDEs ($Hg^{II}$ deposition, photochemical reduction, $Hg^0_{gas}$ re-emission; Ferrari et al., 2005). We propose that in addition to relatively weak and infrequent AMDE activity and Hg deposition in this interior arctic tundra, rapid photochemical re-emission losses of Hg following AMDEs render these events relatively unimportant as a deposition source of Hg. We provided support for this notion using stable Hg isotope analysis in soils from this site in Obrist et al. (2017) which showed that atmospheric $Hg^0_{gas}$ is the dominant source to the interior tundra snowpack.

**4.2 Spatial and temporal patterns of snowbound mercury in the interior arctic snowpack**

Concentrations of $Hg_{tot}$ and $Hg_{diss}$ were measured in the snowpack overlying a tundra ecosystem at Toolik, a snowpack over the adjacent frozen Toolik Lake, and the tundra snowpack along a 170 km transect between Toolik and the Arctic Ocean (Fig. 2, Fig. 7, and Table S1). Total Hg concentrations in all snow samples collected (i.e., tundra snowpack, lake snowpack, and surface snow) were always much higher than $Hg_{diss}$ levels, likely due to impurities and deposition of Hg associated with plant detritus or soil dust. This also resulted in much higher variability of $Hg_{tot}$ concentrations compared to $Hg_{diss}$ concentrations. Due to the high variability in $Hg_{tot}$ concentrations we focused our discussions on $Hg_{diss}$ data. The measurements performed at Toolik showed low levels compared to many other high latitude studies, with $Hg_{diss}$ concentrations averaging 0.17 ng L$^{-1}$ and ranging between 0.08 and 1.15 ng L$^{-1}$, which is at the low end of concentration ranges reported in other studies mainly focused along the coastal zone (0.14–820 ng L$^{-1}$ for both $Hg_{diss}$ and $Hg_{tot}$; Douglas et al., 2005; Douglas and Sturm, 2004; Ferrari et al., 2004, 2005; Kirk et al., 2006; Nerentorp Mastromonaco et al., 2016; St. Louis et al., 2005; Steffen et al., 2002). The low concentrations we measured result in very small pool sizes of $Hg_{diss}$ stored in the snowpack during wintertime compared to temperate studies (Pearson et al., 2015). At Toolik, snowpack pool sizes amounted to 26.9 and 19.7 ng m$^{-2}$ during peak snowpack and prior to the onset of snowmelt in 2014–2015 and 2015–2016, respectively. Most other Arctic studies were performed close to the coast (i.e., Alert and Barrow), and few studies include inland sites such as Toolik (about 200 km south of the Arctic Ocean). Measurements performed along the transect across the Alaska North Slope (from Toolik to the Arctic Coast) in March 2016 indicated $Hg_{diss}$ concentrations decreased significantly from the Arctic Ocean coast to inland (**Figure 7**). This is consistent with previous observations in Alaska in springtime that suggested an ocean influence on the presence of halogens in the coastal atmosphere resulting in higher Hg deposition (Douglas and Sturm, 2004; Landers et al., 1995; Snyder-Conn et al., 1997). We propose that low snowpack Hg concentrations (<0.5 ng L$^{-1}$ for $Hg_{diss}$) are common in inland northern Alaska areas and that the interior arctic snowpacks exhibit lower levels compared to coastal locations that are subjected to a more significant ocean influences and impacts by AMDEs.

To our surprise, no consistent vertical patterns of $Hg_{diss}$ concentrations were found in the snowpack at Toolik (Fig. 2) and along the 200 km transect to the Arctic Coast (Fig. 7). Thus, upper snowpack $Hg_{diss}$ concentrations were not significantly different from those in the deeper layers, which is in contrast to patterns observed in the Sierra Nevada snowpack where strong concentration enhancements (i.e., more than 2-times the average snowpack concentrations (Faïn et al., 2011) were observed in the top 3 cm of the snowpack. This is likely due to lower atmospheric dry deposition inputs in this remote tundra atmosphere compared to more industrialized lower latitudes. Indeed, seasonal measurements at Toolik indicate a generic lack of atmospheric gaseous $Hg^{II}$ during most of the year and very low amounts of total $Hg^{II}$ deposition, i.e., wet, aerosols, plus gaseous $Hg^{II}$ (Obrist et al., 2017). The lack of significant $Hg^{II}$ dry deposition would prevent a Hg enhancement in surface snow and also is consistent with the low pool sizes of Hg in this tundra snowpack. Further support of this notion also includes that fresh snow collected at the surface throughout the arctic winter and spring was not statistically different from snow Hg concentrations contained in the entire snowpack ($0.26 \pm 0.26$ vs $0.17 \pm 0.10$ ng $L^{-1}$, respectively). Yet, another factor to explain a lack of depth gradients in snow Hg concentrations may include that snow layers can be continuously mixed and redistributed by wind gust (e.g., $>5$ m s$^{-1}$ 12% of the time) across the landscape in the Arctic (Cherry et al., 2014).

We observed significant differences in the amount of Hg contained in the snowpacks over the tundra and over the adjacent Toolik Lake. In fact, while $Hg_{diss}$ concentrations were not statistically significantly different between tundra and lake snow (Table S1), snowpack $Hg_{diss}$ loads on the frozen lake were much lower ($6.2 \pm 0.2$ ng m$^{-2}$), i.e., only about ¼, compared to snowpack $Hg_{diss}$ load on the adjacent tundra ($23.3 \pm 5.0$ ng m$^{-2}$). Two reasons may explain the large discrepancy between lake and tundra snowpack Hg loads: (1) lake would not accumulate the snowpack on open water prior to lake freezing in the early fall; (2) low surface roughness over the lake may prevent settling of snowfall and facilitate remobilization of snow by wind transport (Essery et al., 1999; Essery and Pomeroy, 2004). The implication of this process is a reduction of direct atmospheric deposition over Arctic lakes which is consistent with studies that estimated that annual Hg contribution to lakes via wet deposition is small, generally less than 20% of total deposition (Fitzgerald et al., 2005, 2014). Such spatial redistribution of snow across the tundra landscape further implies that both wet deposition and snow accumulation rates are variable leading to spatial heterogeneity of snowmelt Hg inputs.

Little temporal variation in snowpack Hg concentrations was observed between the early season snowpack evolving mainly under darkness and the late season snowpack exposed to solar radiation (Fig. 2 and Fig. 6) although some temporal differences were evident during March and April when AMDEs were present in the region. Snowpack $Hg_{diss}$ concentrations averaged $0.16$ ng $L^{-1}$ both during the completely dark period (i.e., December and January) and after March 1$^{st}$. Such patterns support measurements of $Hg^{0}_{gas}$ throughout the winter that indicated the snowpack to be a relatively inert matrix with little oxidoreduction reactions (oxidation of $Hg^{0}_{gas}$ or reduction of $Hg^{II}$). An apparent trend in surface snow, however, emerged during springtime when $Hg_{diss}$ concentrations reaching $1.15$ ng $L^{-1}$ were temporally measured in surface layers (Fig. 6) in April 2016. This was a period when AMDEs occurred at this site, as evident by depletions of atmospheric $Hg^{0}_{gas}$ with formation and deposition of oxidized atmospheric $Hg^{II}$ (Obrist et al., 2017; Van Dam et al., 2013). Surface snow Hg concentration enhancements during AMDEs are commonly reported in polar regions, with at times Hg concentration enhancements up to

100 times the base concentration in the Arctic (Lalonde et al., 2002; Lindberg et al., 1998; Nerentorp Mastromonaco et al., 2016; Poulain et al., 2004; Steffen et al., 2002). The presence of AMDEs generally results in increased deposition of Hg to snow and ice surfaces, yet such additional deposition often is short-lived due to the photochemical re-emission of $Hg^0_{gas}$ (Kirk et al., 2006). In our study, snow $Hg_{diss}$ in surface snow quickly declined to levels similar to what was observed prior to AMDEs, and no corresponding concentration enhancements were observed deeper in the snowpack. The influence on snow Hg concentrations was therefore small compared to most studies reporting snow Hg enhancement during AMDEs. We attribute this to the large distance to the coast from our study site and the scarcity of AMDEs—and $O_3$ depletion events—that occur at this inland arctic location (Van Dam et al., 2013). Hence, minor impacts of AMDEs can be present in the interior arctic tundra as evident by patterns observed at Toolik some 200 km south of the Arctic Ocean.

Fresh surface snow that was collected throughout the season can serve as a proxy for atmospheric wet deposition Hg concentrations and loads (Faïn et al., 2011). Both low concentrations measured in fresh surface snow as well as low pool sizes as discussed above suggest low wet deposition rates during winter at our inland arctic sites. Estimation of deposition loads by snow collection can be compromised by quick re-volatilization losses of Hg from fresh snowfall (within the first few hours, e.g., Faïn et al., 2013), or snowmelt losses, but we do not consider these processes to be important at this site. The low $Hg_{diss}$ concentrations measured in surface snow ($0.26 \pm 0.26$ ng $L^{-1}$) be lower than the 10th percentile of wet deposition Hg concentrations reported for Kodiak Island in Alaska during the same period (National Atmospheric Deposition Program, 2017). For comparison, snowfall $Hg_{diss}$ concentrations measured at Alert were between 100 and 200 times higher than in our measurements (A. Steffen, personal communication). Using median concentrations in the surface snow multiplied by the amount of wet deposition for each snow-covered season, we estimated the $Hg_{diss}$ load annually deposited by snowfall to 41.3 and 15.3 ng $m^{-2}$ in the 2014–2015 and 2015–2016, respectively. This is up to 100 times lower than values recently provided from a coastal location 400 km northwest of our study site (Douglas et al., 2017) and up to 200 times lower than long-term measurements from Alert between 1998 and 2010 (A. Steffen, personal communication).

**4.31.1 Origin of mercury in the interior arctic snowpack**

Major cation ($Ca^{2+}$, $K^+$, $Mg^{2+}$, $Na^+$, and $NH_4^+$) and anion ($Cl^-$, $NO_3^-$, and $SO_4^{2-}$) values are used to assess the chemical composition and potential origins for Hg in the snowpack (Pearson et al., 2015; **Table 1**). Three main sources of major ions have been identified in snow deposition in North America by several authors (de Caritat et al., 2005; Krnavek et al., 2012; Poulain et al., 2004; Toom Sauntry and Barrie, 2002): (1) marine with sea spray (associated with $Na^+$ and $Cl^-$); (2) lithogenic with rock and soil dust (associated with $Ca^{2+}$ and $Mg^{2+}$); and (3) anthropogenic with long-range acid pollution (associated with $NH_4^+$ and $SO_4^{2-}$). Results of correlation matrices (**Table 2b**) indicated that $Hg_{diss}$ in the fresh surface snow (i.e., top 3 cm) originated from a mix of natural sources, possibly linked to both mineral dust ($Ca^{2+}$) and sea spray ($Cl^-$). The correlation between $Ca^{2+}$ and $Cl^-$ ($\rho = 0.69$), as well as the absence of correlation between $Hg_{diss}$ and $Na^+$ ($\rho = 0.30$), in surface snow samples likely indicated that a part of $Cl^-$ was originated from mineral dust as $CaCl_2$. A minor influence of sea salt was consistent with coastal observations that showed the highest Hg concentrations close to the Arctic Ocean related to bromine

chemistry (Fig. 7; Douglas and Sturm, 2004). In addition, local or regional dust from rock and soil weathering contributed to the wintertime Hg deposition, particularly at the interior sites close to the Brooks Range where higher snow pH reported were from mineral dust that contained carbonates (Douglas and Sturm, 2004). Indeed, the mountain influence was dominant during the two snow-covered seasons at Toolik: 50% of snow events and 80% of dry periods (i.e., periods without snowfall, 90% of

5   the time) coming from the south. Finally, the ternary diagram of major ions showed the highest $Hg_{diss}$ concentrations were measured in snow with the lowest $SO_4^{2-}$ (Fig. 8), indicating that anthropogenic influence from combustion processes was minor or absent for snow Hg deposition. In fact, Alaska generally showed the lowest $SO_4^{2-}$ concentrations among Arctic sites (de Caritat et al., 2005). Norman et al. (1999) also reported relatively small contributions of anthropogenic $SO_4^{2-}$ in snow at Alert (Canada). From this, we propose that the Hg sources in the arctic snowpack is mainly derived from local lithological erosion,

10  and that Arctic Ocean sources are minor contributions.

The lack of consistent statistically significant associations between major ions and $Hg_{diss}$ across the entire snowpack depth (Table 2a) suggests that the original snowfall Hg content was maintained and largely unaltered after deposition, with no clear accumulation or depletion zones as found in other snowpacks (Ferrari et al., 2005; Poulain et al., 2004; Steffen et al., 2014). We found a small, relative enrichment of alkaline earth elements in snowpack samples compared to surface snow, which

15  indicates some additional contributions of local mineral dust, yet this did not result in a measurable increase in snowpack Hg levels. Hence, we suggest no significant additional deposition of Hg (e.g., by dry deposition of gaseous or particulate Hg) to exposed older snow consistent with the lack of correlation to pollution tracers ($SO_4^{2-}$ and $NO_3^-$). We also suggest largely an absence of re-emission losses or elution losses from snow-melt as occurs in temperate snowpacks (discussed in Faïn et al. (2013) and Pearson et al. (2015)). Elution losses are unlikely given that no temperatures above freezing were present in the

20  Arctic until May, and atmospheric re-emissions losses of volatile $Hg^0_{gas}$ were not important in this arctic snowpack for most of the season as discussed above.

Water stable isotope signatures in wet deposition indicated that some of the highest concentrations of $Hg_{diss}$ (April 2016) were present when snow showed consistent values of $\delta^2H$ and $\delta^{18}O$ averaging −245.8 and −32.0‰, respectively (Fig. 9). However, similar water isotope signatures were found also when $Hg_{diss}$ concentrations were lower (e.g., November and December 2015),

25  so that no significant associations of isotope signatures with Hg concentrations were found. Neither the origin of precipitation as shown by the wide range of stable isotope ratios, nor the physical conditions that often causescause isotopic variation in precipitation (e.g., air temperatures that explain up to 50% of isotopic values via mass effects (: Siegenthaler and Oeschger, 1980)), apparently), shaped the Hg concentrations measured in snowthe snowpack.

**54 Conclusions**

30  In this study, we investigated snow Hg dynamics in the interior arctic tundra at Toolik Field Station, Alaska, simultaneously analyzing Hg in: (1) the gas-phase ($Hg^0_{gas}$) of the atmosphere, interstitial snowpack, and soil pores; and (2) in the solid phase in snow ($Hg_{tot}$ and $Hg_{diss}$). Gaseous $Hg^0$ in the atmosphere–snowpack–soil continuum showed consistent concentration patterns

throughout most of the snow season,  with the arctic tundra soil serving as a continuous sink for $Hg^0_{gas}$, important to consider in Arctic Hg cycling. To our surprise, photochemical formation of $Hg^0_{gas}$ in the snowpack was largely absent and played a  minor role in the interior tundra largely limited to periods of active AMDEs. These observations are in contrast with strong photochemical formation of $Hg^0_{gas}$ in surface snow observed at temperate sites and along the arctic coast, resulting in significant photochemical losses of $Hg^0_{gas}$ from these snowpacks. This calls for a regional adjustment of photochemical $Hg^0_{gas}$ losses from the snowpack in models, which should have different treatment for the arctic snowpack compared to temperate snowpacks. Small $Hg_{diss}$ enhancements were temporarily observed in surface snow during springtime, when AMDEs were present, reflecting the typical sequence of Hg deposition to the top snowpack followed by fast photochemical volatilization losses of $Hg^0_{gas}$ during that time. At this interior arctic site, AMDEs, however, resulted in negligible deposition loads. Low concentrations of both $Hg_{tot}$ and $Hg_{diss}$ were measured in the snowpack across this northern Alaska region, resulting in a small reservoir of Hg stored in this snowpack available for potential mobilization during snowmelt ($<30$ ng m$^{-2}$ for $Hg_{diss}$). These low values suggest that wet Hg deposition via snow is not a major source of Hg to this interior arctic site, a notion we previously supported by direct measurements and stable Hg isotopes that showed that two thirds of the Hg source are derived from $Hg^0_{gas}$ deposition. Multielement analysis of  surface snow (top 3 cm) indicated that arctic snowpack Hg originated from a mix of diffuse and likely natural sources, including local mineral dust (associated with $Ca^{2+}$ and $Mg^{2+}$) and, to a lesser extent, regional marine sea spray (associated with $Cl^-$ and $Na^+$).

**Acknowledgements**

We thank Toolik Field Station staff for their support in this project over two years, especially Jeb Timm, Joe Franish, and Faye Ethridge, for helping  with snow collection. We also thank Martin Jiskra (Geosciences Environnement Toulouse) and Christine Olson (DRI) for their field support, Christopher Pearson, Olivia Dillon, and Jacob Hoberg (DRI) for their support with laboratory analyses, and Dominique Colegrove and Tim Molnar (University of Colorado) for helping with field work and data processing. We finally thank Alexandra Steffen for providing mercury snow data from Alert. Funding was provided by the U.S. National Science Foundation (NSF) under award (#PLR 1304305) and cooperative agreement from National Aeronautics and Space Administration (NASA EPSCoR NNX14AN24A).

[revised manuscript text omitted]

---

## Author Response (AR2)

**REVIEWER 2**

The authors did a good job at addressing the reviewers' comments. The manuscript is now easier to follow and the key messages better presented. I only have very minor comments at this point.

page 4, line 17: "concentration data showed here were adjusted". Please very briefly (in one sentence) explain how.

We clarified this point.
The comparison was performed between the two atmospheric inlets (110 cm from the snow tower and 360 cm from the micrometeorological tower), we thus evaluated the deviation across time and corrected this deviation for a better comparison.

page 4: while I understand that lines 7-8 refer to the snow towers, I am not sure I understand what lines 17-19 refer to.

This refers to the snow tower as well. This section describes the sampling sequence and the measurements.

page 8, line 4-6: Angot et al., 2016b refers to Antarctic rather than arctic snowpack measurements.

We corrected that. Thanks!

page 8, line 26: "snowfall amounts at Toolik were much lower than in temperature snowpack". Please provide data or a reference to support this.

We added references.

page 12, line 19: Nerentorp Mastromonaco et al., 2016 refers to AMDEs in Antarctica, not in the Arctic.

Thanks! This is now corrected.

page 12, line 26: "observed in deeper in the snowpack" should be "observed deeper in the snowpack".

Done.

page 14, line 22: "may influenced" should be "may influence".

Done.